# Nicotine dependence among critically ill COVID-19 patients: A population-based cohort study

**John Garza** ⓘ *, **Roy Sebastian** ⓘ, **Brinkley Cover, Asley Sanchez, Vani Selvan**

Texas Tech University Health Sciences Center at the Permian Basin, Odessa, Texas, United States of America

* gar31703@ttuhsc.edu

## Abstract

Nicotine dependence is associated with lower life expectancy and patients who use nicotine are at an increased risk of diseases of the respiratory system. Data on the prognostic impact of nicotine dependence in COVID-19 is inconclusive. We used publicly available, statewide, and deidentified records to study hospitalizations of patients aged ≥ 18 years with a principal diagnosis of COVID-19 admitted to the intensive care units of acute care hospitals in Texas from Q2 2020 through Q4 2024. We measured the effect of nicotine dependence on mortality in the overlap population using weighted multivariable logistic regression and g-computation. We identified 142,045 hospitalizations admitted to intensive care units with a principal diagnosis of COVID-19 of which 10,452 (7.3%) were classified as currently nicotine dependent, 23,671 (16.7%) as formerly nicotine dependent, and 107,922 (76.0%) as never nicotine dependent. Currently and formerly nicotine dependent individuals had significantly higher burden of comorbidities compared to never nicotine dependent individuals; mean [standard deviation] Deyo comorbidity index 1.8 [1.9] and 1.8 [1.9] *vs* 1.4 [1.8] but lower rates of in-hospital mortality 10.7% and 13.8% *vs* 16.1%. On adjusted analysis, current nicotine dependence and former nicotine dependence remained associated with reduced in-hospital and short-term mortality. Additional investigations exploring the mechanisms leading to these findings are necessary.

## Introduction

Severe acute respiratory syndrome coronavirus 2 (SARS-COV-2) is a novel coronavirus that has spread across the globe and resulted in a pandemic [1]. Healthcare systems became overwhelmed with patients, particularly with increased admissions to intensive care units (ICU). Globally, millions were diagnosed with SARS-COV-2 virus and the World Health Organization declared a public health emergency from 30 January 2020–4 May 2023 [2]. The SARS-COV-2 virus spread rapidly, constantly

**Data availability statement:** The data used in this project came from the Texas Inpatient Public Use Data File, 2020-2024, Texas Health Care Information Collection Center for Health Statistics, Texas Department of State Health Services, Austin TX. The data use agreement signed by the authors prevents them from sharing the data themselves. The data is publicly available after completion of a data use agreement form at https://www.dshs.texas.gov/center-health-statistics/texas-health-care-information-collection/download-and-purchase-data/texas-inpatient-public-use-data-file-pudf.

**Funding:** The author(s) received no specific funding for this work.

**Competing interests:** The authors have declared that no competing interests exist.

evolved, leading to widespread coronavirus disease 2019 (COVID-19), and posed a severe public health threat.

SARS-COV-2 primarily affects the epithelial cells within the lungs with the exact knowledge of the mechanism of lung injury remaining unresolved. Mechanisms for lung injury secondary to SARS-COV-2 may be multifactorial. The virus has the capability to enter macrophages and dendritic cells and induce inflammation with pro-inflammatory cytokines and chemokines potentially playing a role [3,4].

Several studies assessed risk factors for the progression and severity of SARS-COV-2 [5]. Many studies highlighted the risk of nicotine use and dependence for increased hospitalization and mortality when compared to never nicotine dependence [6,7]. Similar to other risk factors, nicotine dependence was considered to increase both the risk of contracting SARS-COV-2 as well as the risk of suffering from more complications following infection [8]. Components in tobacco smoke, such as acrolein, formaldehyde, nitrogen oxides, cadmium, and hydrogen cyanide, cause respiratory cilia toxicity, irritants, and oxidant activity [9]. When the oxidative metabolism of cells is altered, their immune systems are weakened which increases the risk of contracting SARS-COV-2 [10].

Nicotine dependence has been associated with high mortality secondary to co-morbid conditions. According to Conklin *et al*, due to the pathophysiological process of tobacco products and nicotine impacts, nicotine users have co-morbid conditions including hypertension (HTN), atherosclerosis, cardiovascular disease (CVD), diabetes type II (DM II), chronic lung disease (CLD), inflammation, vascular injury, and autonomic nervous (ANS) imbalances [11]. Nicotine users with co-morbid conditions like HTN, CVD, and DMII may have worse outcomes than persons without co-morbid conditions [12]. Recent meta-analyses by Gallus *et al* [13] and Tazaerji *et al* [14] concluded that the higher risk among smokers may be attributed to existing co-morbid conditions secondary to smoking.

Although research on COVID-19 outcomes has used the term "smoking" as an exposure, the term "nicotine dependence" is a medically all-encompassing term to describe someone who uses nicotine, irrespective of the delivery method [15,16]. While "smoking" specifically refers to the tobacco that is burnt, "nicotine dependence" may categorize the substance that drives the underlying disease and behavior. "Nicotine dependence," in place of "current smoker," broadens the scope of delivery, covering a wide variety of products that do not involve traditional smoking but still involve nicotine including e-cigarettes, smokeless tobacco – snuff, dip, or chewing tobacco; nicotine replacement therapy- nicotine patches, gums, lozenges, and sprays; and heating tobacco products- where devices are used to heat tobacco without burning, describing nicotine use more as a dependency versus behavior.

"Nicotine dependence" highlights the physiological dependence on the chemical rather than just the act of smoking which provides more medical accuracy in health care and research. "Nicotine dependence" is sometimes used interchangeably for smoking as a behavior. "Nicotine dependence" allows us to broaden the discussion based on whether they are inhaling smoke or using a cleaner nicotine delivery method, like a patch. Despite this, "smoker" remains in use in both medical research, clinical settings, and public health policy.

The scientific literature, facilitating the goals of tobacco science towards producing new knowledge and improving health among individuals, communities, and population levels, continues to use the term "smoker". In today's context of various substance use, referring to someone as a "smoker" is vague and incomplete. One person may be thinking of cigarettes, while another, particularly among adolescents and young adults, may interpret it as another substance, unrelated to cigarettes.

Smoking as an exposure does not require individuals to be dependent on nicotine and smoking includes nontobacco substances such as cannabis that do not contain nicotine. Despite a significant intersection between smoking and nicotine dependence, they are different with neither exposure being a subset of the other. This complicates comparisons of studies using smoking as an exposure with those using nicotine dependence as an exposure in studies of both COVID-19 and other diseases.

In this study, we challenged the understanding of the association of nicotine dependence and mortality among hospitalized patients with a diagnosis of COVID-19 by conducting a retrospective and population level cohort study of all hospitalized patients in Texas with a principal diagnosis of COVID-19 admitted to the ICU at acute care hospitals. We initially hypothesized that currently nicotine dependent individuals would have a higher mortality risk than never nicotine dependent individuals. The results were the opposite of our hypothesis.

## Materials and methods

### Study design

We have completed a population-based cohort study using the Texas Inpatient Public Use Data File (TIPUDF) [17]. The study measures the effect in the overlap population of current *vs* never nicotine dependent individuals on in-hospital mortality among critically-ill patients with a principal diagnosis of COVID-19. Fig 1 provides a directed acyclic graph for the variables included in the study. From the directed acyclic graph, it can be seen that the measured effect of nicotine dependence on mortality is composed of the direct effect of nicotine on mortality and the indirect effect through tobacco smoke of nicotine dependence on mortality. These components are not known to be additive so the measured effect is not interpreted as the sum of these components.

### Data sources

The TIPUDF is an administrative dataset provided by the Texas Department of State Health Services that summarizes information from non-federal hospitals including demographics, diagnoses, procedures, and hospital disposition, capturing approximately 97% of hospital discharges in the state. The TIPUDF is a publicly available and deidentified dataset, and the study was determined to be exempt from formal review by the Texas Tech Health Sciences Center Institutional Review Board. Hospitalizations are used as the unit of analysis since the TIPUDF provides hospitalization-level and not patient-level information, preventing the identification of repeated admissions. We have followed the Strengthening the Reporting of Observational Studies in Epidemiology (STROBE) guidelines [18].

### Patients and variables

**Cohort derivation.** Our primary cohort were hospitalized patients aged ≥ 18 years admitted to the ICU of acute care hospitals in Texas with a principal diagnosis of COVID-19 during the period Q2 2020 – Q4 2024. In order to obtain a homogenous cohort where the entire course of hospital stay was observable, we excluded elective admissions, transfers to and from another hospital, and discharges against medical advice. We identified hospitalizations with a diagnosis of COVID-19 based on the presence of the *International Classification of Diseases, Tenth Revision, Clinical Modification* (ICD-10-CM) code U071 in the principal diagnosis column of the TIPUDF [19]. Hospitalizations with ICU admissions were identified based on hospital charges coded for an ICU or a coronary care unit. Fig 2 reviews the cohort derivation.

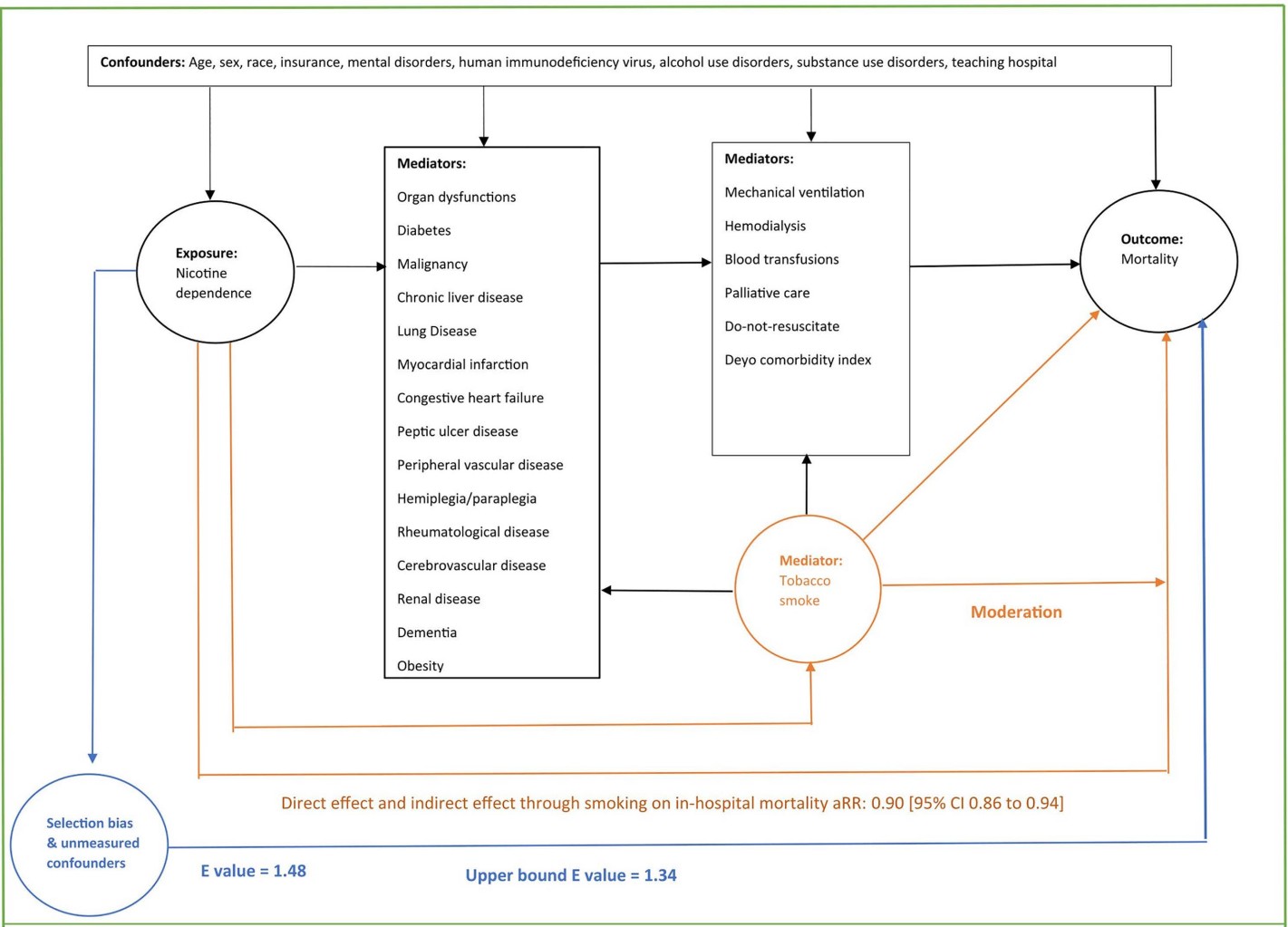

**Fig 1. Directed acyclic graph.**

**Exposure and outcome.** The primary exposure was nicotine dependence. Nicotine dependence is a three-level categorical variable with levels never dependent, formerly dependent, and currently dependent. Fourteen hospitalizations had a diagnosis for both former nicotine dependence and current nicotine dependence. These hospitalizations were classified as currently nicotine dependent. ICD-10-CM codes for nicotine dependence were identified using the 2024 version of the *ICD-10-CM Tabular List of Diseases and Injuries* [19]. Codes for current and former nicotine dependence are listed in S1 Table. S2 Table reports the prevalence of each ICD-10-CM code used to identify former and current nicotine dependence along with code descriptions. From S2 Table, it can be seen that most currently dependent nicotine users were smoking cigarettes. After cigarettes, nicotine dependence unspecified, was the largest portion of the exposure group. According to the *2024 ICD-10-CM Official Guidelines for Coding and Reporting*, unspecified means that the information in the medical record does not allow assigning a more specific code [20]. These hospitalizations may include users of non-tobacco-based nicotine products such as vapes, oral pouches, gums, or patches. However, nicotine dependence unspecified may

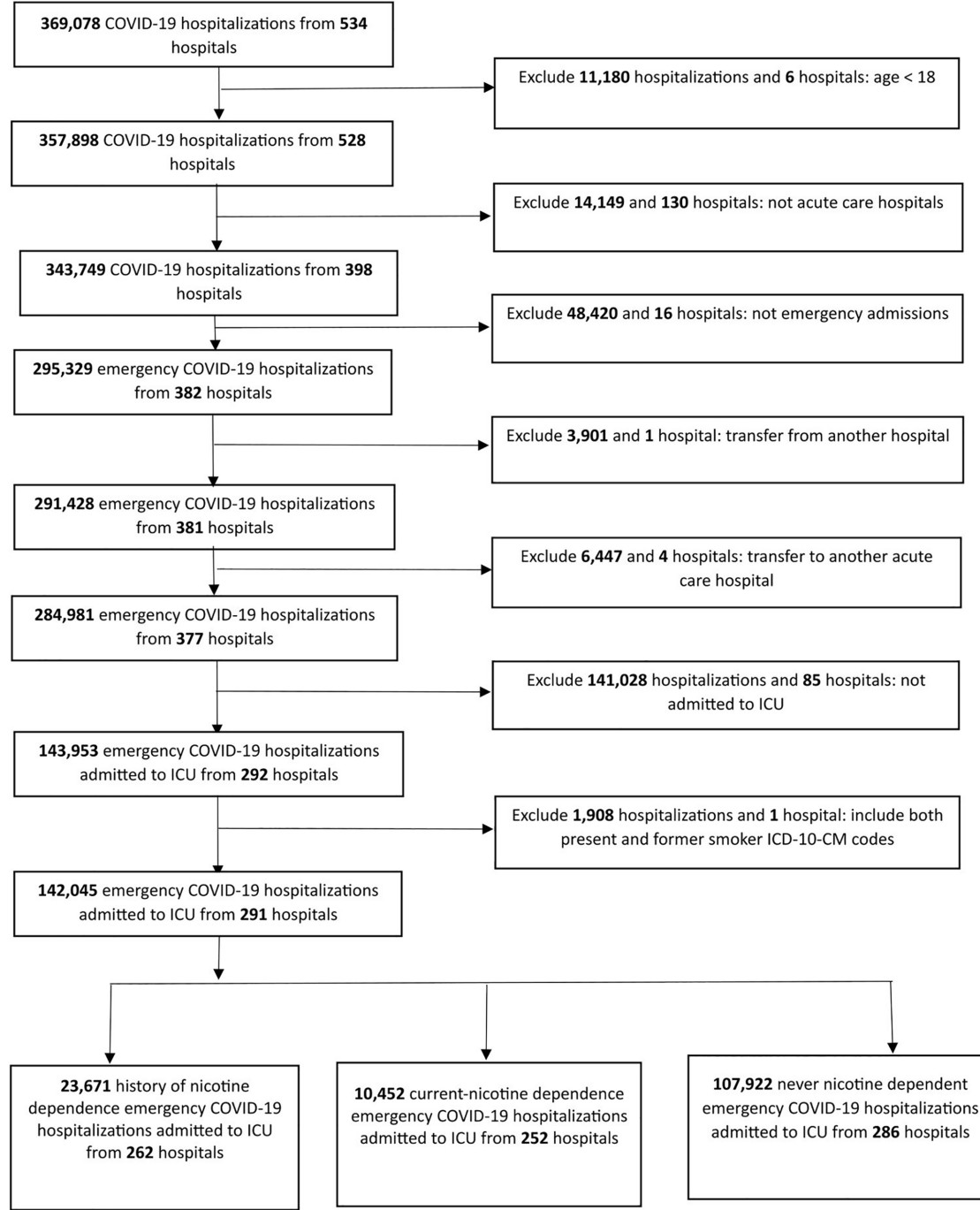

**Fig 2. Cohort derivation.**

also result from a hospital's medical coders not being able to determine from physician notes what the nicotine product used by the patient was and notably may include tobacco users including cigarette, cigar, and pipe smokers. Following cigarettes and unspecified, nicotine dependence, other tobacco product was most identified. These other tobacco

products, include non-cigarette forms of smoking such as cigars and pipes but also include non-smoking products such as tobacco snuff and dissolvable nicotine sticks. Nicotine dependence, chewing tobacco was the least identified class of nicotine dependence.

Codes for former nicotine dependence include codes for patients in withdrawal or remission. Few hospitalizations in the cohort were identified as being in withdrawal or remission. Almost all hospitalizations identified as formerly nicotine dependent were identified using code Z87891 for personal history of nicotine dependence. It is important to understand that this code is used for all forms of nicotine products and consequently there is no set of ICD-10-CM codes that allow identifying only all the former smokers who were previously dependent on nicotine.

We used the Healthcare Cost and Utilizations Projects' *Clinical Classification Software Refined for ICD-10-CM* (CCSR) to search for all other ICD-10-CM codes with either nicotine or tobacco in their code description [21]. A few ICD-10-CM codes not previously mentioned were found. A summary of these codes including code descriptions and prevalence within the study cohort can be seen at the bottom of S2 Table. ICD-10-CM codes Z720 for tobacco use not elsewhere specified, Z716 tobacco abuse counseling, and Z7722 for contact with and (suspected) exposure to environmental tobacco smoke (acute) (chronic) were not used to identify hospitalizations in the exposure group since the *2024 ICD-10-CM Tabular List of Diseases and Injuries* has an excludes 1 note below F17 for Z720, a use additional note below Z716 for F17, and an excludes 1 note for F17 below Z7722. According to the *2024 ICD-10-CM Official Guidelines for Coding and Reporting* an excludes 1 note below a code in the *ICD-10-CM Tabular List of Injuries and Diseases* means that the two codes should not be used together and a use additional note means the codes should appear together [20]. Hospitalizations with Z720 thus identify users of tobacco products including forms of smoking who are not currently dependent on nicotine. Codes O9933x for tobacco use disorder complicating pregnancy, childbirth, and the puerperium were not present on any hospitalization records, code Z5731 for occupational exposure to environmental tobacco smoke was found on only six records, T652x codes for toxic effect of tobacco and nicotine were only found on two hospital records, and P042 does not apply to adult patients.

Although ICD-10-CM codes do not allow a clear partitioning of the cohort into current, former, and never smokers, we have provided a sensitivity analysis for type of nicotine dependence where the reference group excludes all other forms of nicotine dependence and excludes hospitalizations with other codes for tobacco use. These sensitivity analyses, presented in S5 Table includes an analysis for the association of nicotine dependence, cigarettes and mortality. Smoking without reference to nicotine dependence is not a diagnosis captured by ICD-10-CM and several ICD-10-CM codes are ambiguous regarding tobacco smoking. As a result, the use of smoking as an exposure variable in studies based on ICD-10-CM data are complicated by issues with sensitivity, specificity, and the inability to identify former smokers.

The primary outcome was in-hospital mortality. Short-term mortality, defined as in-hospital mortality or discharge to a hospice, was used as a secondary outcome. Results are reported as adjusted risk ratio and 95% confidence interval (aRR [95% CI]).

**Risk-adjustment covariates.** Risk-adjustment covariates were chosen *a priori* and are based on biological and clinical plausibility. Risk-adjustment covariates include demographics (age, gender, race/ethnicity, type of health insurance), major comorbidities (based on the Deyo modification of the Charlson Comorbidity Index [22,23]), alcohol use disorders (CCSR category MBD017), substance use disorders (MBD018 - MBD023, MBD025, and MBD028 - MBD033), obesity (END009), malnutrition (END008), mental disorders (MBD001 – MBD009, and SYM008), hospital teaching status, and year of admission. Mechanical ventilation, hemodialysis, and blood transfusion were used as risk-adjustment covariates and were identified using *International Classification of Diseases, Tenth Revision, Procedure Coding System* (ICD-10-PCS) codes [24]. We used *Clinical Classification Software Refined for ICD-10-PCS v2024.1* to identify ICD-10-PCS codes for mechanical ventilation (ESA003), hemodialysis (ESA001), and blood transfusions (ADM 001 and ADM002) [25]. Palliative care was identified using ICD-10-CM code Z515 from CCSR category *FAC010: Other aftercare encounter.* Do not resuscitate orders were identified using ICD-10-CM code Z66 from CCSR category *FAC025: Other specified status.* Organ dysfunctions were defined using the method of Martin and colleagues [26].

## Statistical analysis

Categorical variables are summarized as counts and percentages. Continuous variables are summarized as mean and standard deviation. The $\chi^2$ test was used for comparison of categorical variables and nonparametric permutational analysis of variance was used for comparison of continuous variables. Two-sided p values < 0.05 were considered statistically significant. The primary analysis procedure was propensity score overlap weighting with weighted multivariable logistic regression and g-computation using the *WeightIt* and *marginaleffects* R packages [27,28].

**Propensity score calculation.** The propensity score is the probability of treatment given the measured covariates. Propensity scores for nicotine dependence were computed using multivariable logistic regression with nicotine dependence used as the dependent variable. All covariates used for risk adjustment were included in the propensity score calculation. Density plots for propensity scores of each pairwise comparison of nicotine dependence status are provided in S1 Fig.

**Overlap weights.** For a given pairwise comparison of nicotine dependence, the overlap weight of an exposed hospitalization is one minus the propensity of exposure for that hospitalization and the propensity of an unexposed hospitalization is the propensity of exposure for that hospitalization [29,30]. Thus, the overlap weight of a hospitalization is the probability of being in the opposite treatment group. For binary exposures, overlap weights result exact balance of covariates used in the propensity score calculation. The overlap weights were computed using the *WeightIt* R package.

**Overlap weighted population.** The overlap weights were normalized to sum to one across the whole cohort and then used as probabilities to construct the overlap weighted population. The overlap weighted population is a pseudo population whose elements are hospitalizations from the original cohort but whose relative frequency in the pseudo population is the normalized overlap weight. The overlap weighted population results in clinical *equipoise* of the exposure variable. Properties of clinical *equipoise* include the following. First, each exposure group is equally represented in the overlap weighted population. Thus, if a hospitalization is randomly drawn from the overlap weighted population, it is equally likely to be in the exposed or unexposed group. Second, each level of each risk-adjustment covariate has the same expected value in the overlap weighted population. This property is referred to as exact balance. Third, in the overlap weighted population, each individual hospitalization has equal probability of being in the exposed or unexposed group given the measured covariates. This property of clinical *equipoise* applies to each individual hospitalization. Fourth, in the overlap weighted population, the logistic regression coefficients of regression onto the exposure variable are all zero, meaning that there are no arrows directed at the exposure variable for a directed acyclic graph drawn in the overlap weighted population and consequently there are no known confounders in the overlap weighted population. Demonstrations of these properties are provided in the R code in S1 File. Fig 3 displays the absolute standardized difference for covariates before and after overlap weighting for the comparison of current *vs* never nicotine dependence.

**G-computation with weighted multivariable logistic regression.** To increase precision, weighted multivariable logistic regression was applied on the overlap weighted population using the overlap weights with mortality as outcome. Although the point estimates for aRR and aRD can be computed using only the *glm* function from the *stats* R package and consequently nonparametric bootstraps could have been used to obtain all confidence intervals for the study while avoiding use of the *WeightIt* and *marginaleffects* packages, the desire to conduct many sensitivity and subgroup analyses, revise definitions of covariates, and change covariates included for risk adjustment, makes use of nonparametric bootstraps impractical for the study due to the time required to run them. As an alternative, the output of the *weightit* function was supplied to the *glm_weightit* function whose output was supplied to the *avg_comparisons* function of the *marginaleffects* package. This allows for confidence intervals of the risk ratio that apply to the overlap weighted population and that account for both the uncertainty of the propensity score model and the uncertainty of the weighted logistic regression model simultaneously. As a result, the confidence intervals in this study approximate those that would be obtained using nonparametric bootstrapping of the entire analysis procedure without using the *WeightIt* and *marginaleffects* packages. The R code in S1 File includes a comparison of the confidence interval resulting from 10,000 non-parametric bootstraps

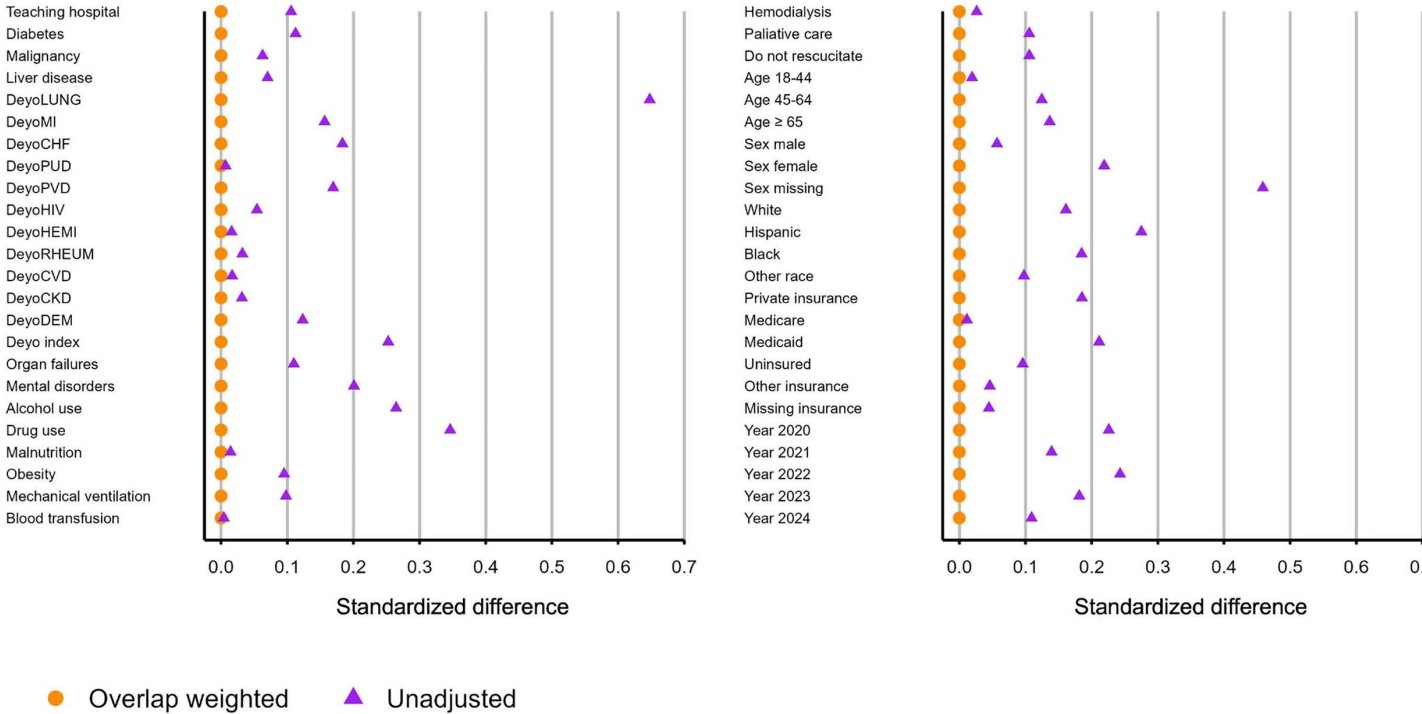

**Fig 3. Covariate balance before and after overlap weighting.**

of the point estimate using only the *glm* function from the *stats* R package with the confidence interval obtained using the described analysis procedure based on the *WeightIt* and *marginaleffect*s packages. The comparison demonstrates that the analysis procedure results in a confidence interval of very similar width to that obtained through nonparametric bootstraps.

## Alternative modeling for the potential impact of multicollinearity

Variance inflation factors for the weighted multivariable logistic regression model are reported in S3 Table. Deyo comorbidity index has variance inflation factor above recommended limits. This is to be expected as the Deyo comorbidity index is a linear combination of the Deyo-Charlson comorbidities. The focal variable, nicotine dependence, does not have variance inflation factor above recommended limits so that the multicollinearity related to Deyo comorbidity index does not automatically invalidate the model. The results of alternative analyses for the evaluation of the potential impact of multicollinearity on the model are presented in S4 Table. The results show that multicollinearity has no material impact on the model results and we have retained the Deyo comorbidity index as a risk adjustment covariate since it accounts for interactions between the Deyo-Charlson comorbidities.

**Assessment of selection bias.** We explored the plausibility of selection and collider bias by applying the primary analysis approach to other comorbidities expected to increase mortality among COVID-19 patients and to mental and substance use disorders [31,32]. The results are presented in S6 Table. An example selection bias could be restriction to hospitalized patients admitted to the ICU as seen in Fig 4 where nicotine dependence and COVID-19 illness severity collide on ICU admission. In this case, restriction to ICU patients could result in an imbalance of COVID-19 severity (which is not measurable with the TIPUDF) between never and currently nicotine dependent individuals.

**Adults with COVID-19**

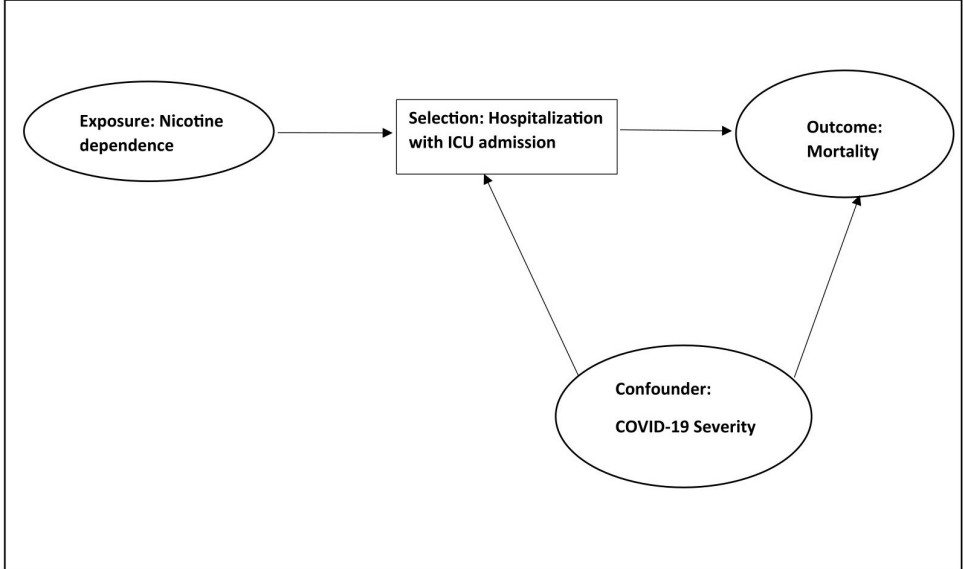

**Fig 4. Possible selection bias.**

An example of possible selection bias is currently nicotine dependent individuals expiring more often before hospital admission leaving less severely ill currently nicotine dependent individuals hospitalized. Another example is currently nicotine dependent individuals being admitted to the ICU with lower severity of COVID-19 due to higher burden of comorbidities. A third example is access to care differences such that severely ill currently nicotine dependent individuals have less hospital access compared to similarly ill never nicotine dependent individuals. If these selection biases are present, then other diagnoses could demonstrate a similar protective association.

**Sensitivity analyses.** We performed six sensitivity analyses. First, we examined the sensitivity of the association of nicotine dependence to mortality when using the secondary outcome of short-term mortality defined as in-hospital mortality or discharge to a hospice. Second, we performed analyses to evaluate the sensitivity of the association between nicotine dependence and mortality to hospitalizations at higher risk of mortality. Cases included age ≥ 65 years, invasive mechanical ventilation, Deyo comorbidity index ≥ 3, and total organ dysfunctions ≥ 3. Third, we considered the sensitivity of the association to year of discharge. Fourth we considered the sensitivity of the association to the ICD-10-CM definition of the exposure. The primary analysis approach was used for these first four sensitivity analyses. Fifth, we computed the E-values for the effect of nicotine dependence on mortality in the overlap weighted population. The E-value is the minimum level of association on the risk ratio scale that unmeasured confounders must have with both the exposure and outcome, beyond the adjusted covariates, to explain away the observed association [33]. Lastly, we have computed bounding factors for selection bias on the risk ratio scale using appropriate formulas for the present case of inference within a sampled population [34].

**Subgroup analyses.** Additional analyses were performed to evaluate the consistency of the association between smoking and hospital mortality among *a priori* selected subgroups. Subgroups examined included age, gender, race/ethnicity, Deyo comorbidity index, and number of organ dysfunctions. The primary analysis approach was used for subgroup analyses. Subgroup analyses were completed for both In-hospital mortality and short-term mortality.

**Software and data management.** Data management used Microsoft Excel (Microsoft, Redmond, Washington) and statistical analyses used R 4.2.2 (R Foundation for Statistical Computing, Vienna, Austria). The R codes used for this study are provided in the supporting materials file.

## Results

A total of 142,045 hospitalizations from 291 hospitals with a principal diagnosis of COVID-19 and admitted to the ICU were included in the study of which 10,452 (7.3%) were currently nicotine dependent, 23,671 (16.7%) were formerly nicotine dependent, and 107,922 (76.0%) were never nicotine dependent. Crude in-hospital mortality was lower among current dependents (10.7%) and former dependents (13.8%) compared to never dependents (16.1%). On adjusted analyses, currently and formerly nicotine dependent individuals remained associated with lower mortality.

### Cohort characteristics

The unweighted characteristics of currently, formerly, and never dependent hospitalizations are summarized in Table 1. Formerly nicotine dependent individuals were older compared to currently and never nicotine dependent individuals (64.4%, 40.6%, and 49.1% aged ≥ 65 years respectively). Currently and formerly nicotine dependent individuals had higher comorbidity index compared to never nicotine dependent individuals (mean [SD] 1.8 [1.9], 1.8[1.9], *vs* 1.4 [1.8] respectively). Never dependents had more frequent need for mechanical ventilation compared to current and former dependents (12.2% *vs* 9.1% and 9.5% respectively); p < 0.0001 for each comparison. The more frequent need for mechanical ventilation may indicate a higher level of illness severity among never dependents.

### Overlap weighted population

**Clinical equipoise.** In the overlap weighted population, each level of each risk-adjustment covariate was exactly balanced. Fig 3 displays the absolute standardized difference in the weighted and unweighted cohorts for each level of each risk adjustment covariate. Table 2 reports the means and rates of each level of each covariate within the overlap weighted population. The aRR for current *vs* never dependents reported in this manuscript is to be interpreted as the remaining effect, after excluding indirect effects through the risk adjustment covariates, within the overlap weighted population having summary statistics presented in Table 2. In this overlap weighted population current *vs* never nicotine dependence is in clinical *equipoise.* Properties of clinical *equipoise* are verifiable within the overlap weighted population. First, the proportions of current and never dependents are one-half in the overlap weighted population which can be observed by noting that sum of normalized overlap weights within each group is one-half. Secondly, as reported in Table 2, each level of each risk adjustment covariate has the same expected value for never and current dependents. Third, as demonstrated in S1 File, within the overlap weighted population, each hospitalization has the same probability of being currently dependent or being never dependent. Fourth as reported in S7 Table and described in S1 File, the coefficients for multivariable logistic regression onto nicotine dependence are zero for all covariates implying that there are no arrows directed into nicotine dependence in a directed acyclic graph of the overlap weighted population so that there are no confounders in the overlap weighted population.

**Effective sample size.** The application of overlap weights results in decreased precision compared to unweighted samples. The effective sample size of the overlap weighted population is the size of an unweighted sample that would result in the same precision, that is width of confidence intervals for point estimates. The effective sample size for never and currently nicotine dependent individuals were computed using the formulas provided by Shook-Sa *et al* [35]. The R code in S1 File includes a demonstration of the effective sample size calculation. The effective sample size, after overlap weighting, for never nicotine dependent individuals is 54,723 compared to the original unweighted sample size of 107,922. The effective sample size for currently nicotine dependent individuals is 9,978 compared to the original unweighted

**Table 1. The relative characteristics and outcomes of critically ill COVID-19 patients partitioned by nicotine dependence status.**

| Variable | Current dependent[a] n = 10,452 | Former dependent[a] n = 23,671 | Never dependent[a] n = 107,922 | p value[b] |
|---|---|---|---|---|
| **Age, years** | | | | 0.0001 |
| 18 - 44 | 1,805 (17.3) | 1,776 (7.5) | 16,431 (15.2) | |
| 45 - 64 | 4,401 (42.1) | 6,647 (28.1) | 38,453 (35.6) | |
| ≥ 65 | 4,246 (40.6) | 15,248 (64.4) | 53,038 (49.1) | |
| **Sex[c]** | | | | 0.0001 |
| Male | 5,078 (48.6) | 14,173 (59.9) | 52,351 (48.5) | |
| Female | 3,384 (32.4) | 8,563 (36.2) | 52,232 (48.4) | |
| Suppressed | 1,990 (19.0) | 935 (3.9) | 3,339 (3.1) | |
| **Race/ethnicity** | | | | 0.0001 |
| White | 5,575 (53.3) | 13,951 (58.9) | 48,450 (44.9) | |
| Hispanic | 2,440 (23.3) | 5,757 (24.3) | 37,731 (35.0) | |
| Black | 1,841 (17.6) | 2,634 (11.1) | 13,289 (12.3) | |
| Other | 596 (5.7) | 1,328 (5.6) | 8,450 (7.8) | |
| Missing | 0 (0.0) | 1 (0.0) | 2 (0.0) | |
| **Health Insurance** | | | | 0.0001 |
| Private | 4,605 (44.1) | 10,599 (44.8) | 53,776 (49.8) | |
| Medicare | 3,054 (29.2) | 9,672 (40.9) | 33,709 (31.2) | |
| Medicaid | 899 (8.6) | 865 (3.7) | 4,511 (4.2) | |
| Uninsured | 987 (9.4) | 1,006 (4.2) | 8,055 (7.5) | |
| Other | 675 (6.5) | 1,226 (5.2) | 6,082 (5.6) | |
| Missing | 232 (2.2) | 303 (1.3) | 1,789 (1.7) | |
| **Deyo comorbidity index** (mean, SD[d]) | 1.83 (1.87) | 1.75 (1.87) | 1.41 (1.80) | 0.0001 |
| **Deyo-Charlson comorbidities** | | | | |
| Diabetes | 4,028 (38.5) | 10,342 (43.7) | 47,675 (44.2) | 0.0001 |
| Chronic lung disease | 4,525 (43.3) | 8,563 (36.2) | 19,015 (17.6) | 0.0001 |
| Congestive heart failure | 2,617 (25.0) | 6,084 (25.7) | 21,220 (19.7) | 0.0001 |
| Myocardial infarction | 1,304 (12.5) | 2,835 (12.0) | 9,218 (8.5) | 0.0001 |
| Renal disease | 1,926 (18.4) | 5,468 (23.1) | 22,308 (20.7) | 0.0001 |
| Dementia | 551 (5.3) | 2,127 (9.0) | 9,396 (8.7) | 0.0001 |
| Chronic liver disease | 618 (5.9) | 978 (4.1) | 4,606 (4.3) | 0.0001 |
| Peripheral vascular disease | 488 (4.7) | 984 (4.2) | 2,265 (2.1) | 0.0001 |
| Any malignancy | 496 (4.7) | 1,475 (6.2) | 4,281 (4.0) | 0.0001 |
| Cerebrovascular disease | 443 (4.2) | 1,000 (4.2) | 4,467 (4.1) | 0.7666 |
| Rheumatological disease | 326 (3.1) | 821 (3.5) | 3,172 (2.9) | 0.0001 |
| Hemiplegia or paraplegia | 104 (1.0) | 189 (0.8) | 1,162 (1.1) | 0.0005 |
| Peptic ulcer disease | 75 (0.7) | 114 (0.5) | 714 (0.7) | 0.0051 |
| HIV | 54 (0.5) | 37 (0.2) | 224 (0.2) | 0.0001 |
| **Other comorbidities** | | | | |
| Obesity | 3,262 (31.2) | 6,972 (29.5) | 36,526 (33.8) | 0.0001 |
| Mental disorders | 2,977 (28.5) | 5,917 (25.0) | 22,262 (20.6) | 0.0001 |
| Drug abuse | 820 (7.8) | 386 (1.6) | 1,404 (1.3) | 0.0001 |
| Malnutrition | 709 (6.8) | 1,358 (5.7) | 7,365 (6.8) | 0.0001 |
| Alcohol abuse | 672 (6.4) | 438 (1.9) | 1,209 (1.1) | 0.0001 |
| **Number of organ dysfunctions** (mean, SD[d]) | 1.33 (0.99) | 1.30 (0.99) | 1.45 (1.06) | 0.0001 |

*(Continued)*

**Table 1.** (Continued)

| Variable | Current dependent[a] | Former dependent[a] | Never dependent[a] | p value[b] |
|---|---|---|---|---|
| | n = 10,452 | n = 23,671 | n = 107,922 | |
| **Types of organ dysfunctions** | | | | |
| Respiratory | 6,973 (66.7) | 16,385 (69.2) | 80,298 (74.4) | 0.0001 |
| Renal | 3,094 (29.6) | 7,465 (31.5) | 36,278 (33.6) | 0.0001 |
| Neurological | 432 (4.1) | 932 (3.9) | 5,231 (4.8) | 0.0001 |
| Cardiovascular | 847 (8.1) | 1,885 (8.0) | 9,309 (8.6) | 0.0022 |
| Metabolic | 1,071 (10.2) | 2,415 (10.2) | 12,895 (11.9) | 0.0001 |
| Hematological | 1,034 (9.9) | 2,266 (9.6) | 11,052 (10.2) | 0.0054 |
| Hepatic | 124 (1.2) | 213 (0.9) | 1,950 (1.8) | 0.0001 |
| **Medical procedures** | | | | |
| Do-not-resuscitate order | 1,239 (11.9) | 4,284 (18.1) | 17,290 (16.0) | 0.0001 |
| Invasive mechanical ventilation | 989 (9.5) | 2,159 (9.1) | 13,142 (12.2) | 0.0001 |
| Palliative care | 663 (6.3) | 2,103 (8.9) | 9,501 (8.8) | 0.0001 |
| Hemodialysis | 429 (4.1) | 1,002 (4.2) | 5,330 (4.9) | 0.0001 |
| Blood transfusion | 476 (4.6) | 1,045 (4.4) | 5,324 (4.9) | 0.0016 |
| **Teaching hospital** | 2,264 (21.7) | 5,061 (21.4) | 22,470 (20.8) | 0.0319 |
| **Weekend admission** | 2,743 (26.2) | 6,117 (25.8) | 28,644 (26.5) | 0.0780 |
| **Length of stay** (mean, SD[d]) | 8.43 (7.95) | 8.18 (7.95) | 9.84 (11.20) | 0.0001 |
| **Hospital disposition** | | | | 0.0001 |
| Chronic care facility[e] | 1,385 (13.3) | 4,223 (17.8) | 17,241 (16.0) | |
| Home, self-care | 6,454 (61.7) | 11,764 (49.7) | 57,476 (53.3) | |
| Home, skilled care | 1,133 (10.8) | 3,255 (13.8) | 11,165 (10.3) | |
| Hospice | 295 (2.8) | 1,024 (4.3) | 4,233 (3.9) | |
| In-hospital mortality | 1,115 (10.7) | 3,256 (13.8) | 17,385 (16.1) | |
| Other | 70 (0.7) | 149 (0.6) | 422 (0.4) | |

[a] Parenthesized figures represent percentages except for Deyo comorbidity index, number of organ failures, and length of stay.

[b] Two-sided p values using analysis of variance for continuous variables and $\chi^2$ test for categorical variables.

[c] The State of Texas suppresses the gender data of hospitalization with a diagnosis of HIV, drug abuse, or alcohol abuse.

[d] SD: Standard deviation.

[e] Chronic care facilities include long-term care hospitals, inpatient rehabilitation, skilled nursing facilities, and nursing homes.

sample size of 10,452. The use of overlap weights has resulted in an overlap weighted population that retains significant effective sample size.

## Association of nicotine dependence and in-hospital mortality

**Main results.** Crude in-hospital mortality was 10.7% for currently nicotine dependent individuals, 13.8% for formerly nicotine dependent individuals, and 16.1% for never nicotine dependent individuals. The results of the primary analyses are reported in Table 3. All pairwise contrasts of nicotine dependence status were statistically significant for both in-hospital and short-term mortality. Within the overlap weighted population for current vs never dependents, in-hospital mortality remained lower for currently nicotine dependent individuals 11.4% vs 12.7%.

**Sensitivity analyses for higher risk of mortality.** Sensitivity analyses for hospitalizations at higher risk of mortality are presented in Table 4. The associations of current vs never dependent with mortality were significant across all examined strata, for both in-hospital and short-term mortality, and on both unadjusted and adjusted analysis. These sensitivity

**Table 2. Overlap weighted means of COVID-19 hospitalizations for current and never nicotine dependent individuals.**

| Variables | Never dependent individuals | Currently dependent individuals |
|---|---|---|
| | P = 0.5000 | P = 0.5000 |
| **Age, years (%)** | | |
| 18-44 | 17.7 | 17.7 |
| 45-64 | 40.6 | 40.6 |
| ≥ 65 | 41.7 | 41.7 |
| **Gender (%)** | | |
| Male | 52.0 | 52.0 |
| Female | 35.2 | 35.2 |
| Suppressed | 12.8 | 12.8 |
| **Race/ethnicity (%)** | | |
| White | 51.9 | 51.9 |
| Hispanic | 25.4 | 25.4 |
| Black | 16.6 | 16.6 |
| Other Race | 6.1 | 6.1 |
| **Insurance (%)** | | |
| Private insurance | 45.4 | 45.4 |
| Medicare | 29.7 | 29.7 |
| Medicaid | 7.3 | 7.3 |
| Uninsured | 9.0 | 9.0 |
| Other insurance | 6.4 | 6.4 |
| Missing insurance | 2.2 | 2.2 |
| **Deyo index (mean)** | 1.67 | 1.67 |
| **Deyo-Charlson comorbidities (%)** | | |
| Diabetes | 39.9 | 39.9 |
| Malignancy | 4.6 | 4.6 |
| Chronic liver disease | 5.5 | 5.5 |
| Lung disease | 38.2 | 38.2 |
| MI | 11.4 | 11.4 |
| CHF | 23.5 | 23.5 |
| Peptic ulcer disease | 0.7 | 0.7 |
| Peripheral vascular disease | 4.0 | 4.0 |
| HIV | 0.5 | 0.5 |
| Hemiplegia/paraplegia | 1.0 | 1.0 |
| Rheumatological diseases | 3.1 | 3.1 |
| Cerebrovascular disease | 4.2 | 4.2 |
| Renal disease | 18.8 | 18.8 |
| Dementia | 5.7 | 5.7 |
| **Other comorbidities (%)** | | |
| Mental disorders | 26.9 | 26.9 |
| Alcohol use disorders | 4.4 | 4.4 |
| Drug use disorders | 5.1 | 5.1 |
| Malnutrition | 6.5 | 6.5 |
| Obesity | 32.5 | 32.5 |

*(Continued)*

**Table 2.** (Continued)

| Variables | Never dependent individuals | Currently dependent individuals |
|---|---|---|
| | P = 0.5000 | P = 0.5000 |
| **Medical procedures (%)** | | |
| Mechanical ventilation | 9.9 | 9.9 |
| Blood transfusions | 4.7 | 4.7 |
| Hemodialysis | 4.3 | 4.3 |
| Palliative care | 6.5 | 6.5 |
| Do not resuscitate | 12.4 | 12.4 |
| **Year of discharge (%)** | | |
| Year 2020 | 22.4 | 22.4 |
| Year 2021 | 41.6 | 41.6 |
| Year 2022 | 21.9 | 21.9 |
| Year 2023 | 8.6 | 8.6 |
| Year 2024 | 5.5 | 5.5 |
| **Teaching hospital (%)** | 21.3 | 21.3 |
| **Organ dysfunctions (mean)** | 1.33 | 1.33 |
| **Mortality** | | |
| In-hospital (%) | 12.7 | 11.4 |
| Short-term (%) | 16.0 | 14.3 |

**Table 3. The association of nicotine dependence and mortality among critically-ill COVID-19 patients.**

| Contrast | Numerator | Denominator | aRR [95% CI][a] | aRD [95% CI][b] | p value |
|---|---|---|---|---|---|
| | mortalities/ total no. (%) | | | | |
| **Current dependent *vs* never dependent** | | | | | |
| In-hospital mortality | 1,115/ 10,452 (10.7) | 17,385/ 107,922 (16.1) | 0.8955 [0.8572 to 0.9356] | −0.0134 [−0.0185 to −0.0083] | < 0.0001 |
| Short-term mortality | 1,410/ 10,452 (13.5) | 21,618/ 107,922 (20.0) | 0.8926 [0.8618 to 0.9245] | −0.0174 [−0.0225 to −0.0122] | < 0.0001 |
| **Current dependent *vs* former dependent** | | | | | |
| In-hospital mortality | 1,115/ 10,452 (10.7) | 3,256/ 23,671 (13.8) | 0.9432 [0.8977 to 0.9911] | −0.0069 [−0.0127 to −0.0011] | 0.0207 |
| Short-term mortality | 1,410/ 10,452 (13.5) | 4,280/ 23,671 (18.1) | 0.9438 [0.9073 to 0.9818] | −0.0088 [−0.0147 to −0.0029] | 0.0041 |
| **Former dependent *vs* never dependent** | | | | | |
| In-hospital mortality | 3,256/ 23,671 (13.8) | 17,385/ 107,922 (16.1) | 0.9388 [0.9149 to 0.9633] | −0.0093 [−0.0130 to −0.0056] | < 0.0001 |
| Short-term mortality | 4,280/ 23,671 (18.1) | 21,618/ 107,922 (20.0) | 0.9477 [0.9294 to 0.9664] | −0.0103 [−0.0140 to −0.0066] | < 0.0001 |

[a] aRR [95% CI]: adjusted risk ratio and 95% confidence interval.

[b] aRD [95% CI]: adjusted risk difference and 95% confidence interval.

analyses are important as they show that the association of nicotine dependence and mortality is strong enough to remain present even after restricting analysis to patients at very high risk of death.

**Sensitivity analyses for year of discharge.** Table 5 reports the results of sensitivity analyses to year of discharge for current *vs* never dependents. The unadjusted in-hospital and short-term mortality rates were lower for never dependents in each year. On adjusted analysis, the associations for short-term mortality were consistent for 2020, 2021, 2022, and 2023. For 2024, the adjusted results are inconclusive possibly due to lower power resulting from fewer hospitalizations and lower mortality rates. These sensitivity analyses for year of discharge are important for at least a few reasons. First, earlier in the pandemic, ICU capacity and treatment protocols were different. Second, later treatment strategies with antivirals could have modified the effects of nicotine dependence on mortality. Third, many research studies have used data

**Table 4. Sensitivity analyses for the association of current versus never nicotine dependent and mortality among critically-ill COVID-19 patients.**

| Contrast | Current dependent | Never dependent | aRR [95% CI][a] | aRD [95% CI][b] | p value |
|---|---|---|---|---|---|
| | *mortalities/ total no. (%)* | | | | |
| **Deyo index ≥ 3** | | | | | |
| In-hospital mortality | 348/ 3,163 (11.0) | 4,632/ 23,319 (19.9) | 0.8219 [0.7550 to 0.8948] | −0.0267 [−0.0376 to −0.0159] | < 0.0001 |
| Short-term mortality | 518/ 3,163 (16.4) | 6,466/ 23,319 (27.7) | 0.8514 [0.8013 to 0.9046] | −0.0315 [−0.0428 to −0.0202] | < 0.0001 |
| **Age ≥ 65** | | | | | |
| In-hospital mortality | 519/ 4,246 (12.2) | 10,287/ 53,038 (19.4) | 0.8153 [0.7639 to 0.8702] | −0.0299 [−0.0387 to −0.0211] | < 0.0001 |
| Short-term mortality | 752/ 4,246 (17.7) | 13,963/ 53,038 (26.3) | 0.8418 [0.8029 to 0.8826] | −0.0354 [−0.0445 to −0.0263] | < 0.0001 |
| **Organ dysfunctions ≥ 3** | | | | | |
| In-hospital mortality | 497/ 1,223 (40.6) | 8,180/ 16,182 (50.5) | 0.9282 [0.8762 to 0.9832] | −0.0325 [−0.0569 to −0.0080] | 0.0112 |
| Short-term mortality | 577/ 1,223 (47.2) | 9,406/ 16,182 (58.1) | 0.9318 [0.8894 to 0.9761] | −0.0356 [−0.0584 to −0.0128] | 0.0029 |
| **Invasive mechanical ventilation** | | | | | |
| In-hospital mortality | 553/ 989 (55.9) | 8,882/ 13,142 (67.6) | 0.9436 [0.8985 to 0.9910] | −0.0346 [−0.0632 to −0.0060] | 0.0202 |
| Short-term mortality | 611/ 989 (61.8) | 9,600/ 13,142 (73.0) | 0.9516 [0.9144 to 0.9904] | −0.0323 [−0.0579 to −0.0067] | 0.0150 |

[a] aRR [95% CI]: adjusted risk ratio and 95% confidence interval.

[b] aRD [95% CI]: adjusted risk difference and 95% confidence interval.

**Table 5. Sensitivity analyses by year for the association of nicotine dependence and mortality among critically-ill COVID-19 patients.**

| Subgroup | Current dependent | Never dependent | aRR [95% CI][a] | aRD [95% CI][b] | p value |
|---|---|---|---|---|---|
| | *mortalities/ total no. (%)* | | | | |
| **2020[c]** | | | | | |
| In-hospital mortality | 261/ 2,162 (12.1) | 5,345/ 32,665 (16.4) | 0.9255 [0.8480 to 1.0101] | −0.0102 [−0.0214 to 0.0010] | 0.0827 |
| Short-term mortality | 310/ 2,162 (14.3) | 6,479/ 32,665 (19.8) | 0.9159 [0.8516 to 0.9850] | −0.0137 [−0.0247 to −0.0028] | 0.0179 |
| **2021** | | | | | |
| In-hospital mortality | 631/ 4,156 (15.2) | 9,461/ 48,077 (19.7) | 0.9232 [0.8743 to 0.9747] | −0.0132 [−0.0219 to −0.0045] | 0.0039 |
| Short-term mortality | 736/ 4,156 (17.7) | 11,114/ 48,077 (23.1) | 0.9233 [0.8828 to 0.9657] | −0.0154 [−0.0238 to −0.0070] | 0.0005 |
| **2022** | | | | | |
| In-hospital mortality | 177/ 2,439 (7.3) | 2,071/ 16,636 (12.4) | 0.8086 [0.7137 to 0.9162] | −0.0187 [−0.0290 to −0.0084] | 0.0009 |
| Short-term mortality | 260/ 2,439 (10.7) | 2,973/ 16,636 (17.9) | 0.8026 [0.7334 to 0.8782] | −0.0281 [−0.0389 to −0.0174] | < 0.0001 |
| **2023** | | | | | |
| In-hospital mortality | 29/ 966 (3.0) | 314/ 6,112 (5.1) | 0.6409 [0.4590 to 0.8949] | −0.0175 [−0.0290 to −0.0060] | 0.0090 |
| Short-term mortality | 61/ 966 (6.3) | 632/ 6,112 (10.3) | 0.7658 [0.6199 to 0.9460] | −0.0207 [−0.0359 to −0.0055] | 0.0133 |
| **2024** | | | | | |
| In-hospital mortality | 17/ 729 (2.3) | 194/ 4,432 (4.4) | 0.9544 [0.6127 to 1.4867] | −0.0014 [−0.0145 to 0.0117] | 0.8366 |
| Short-term mortality | 43/ 729 (5.9) | 420/ 4,432 (9.5) | 0.9481 [0.7004 to 1.2834] | −0.0037 [−0.0241 to 0.0168] | 0.7300 |

[a] aRR [95% CI]: adjusted risk ratio and 95% confidence interval.

[b] aRD [95% CI]: adjusted risk difference and 95% confidence interval.

[c] Q2, Q3, Q4 only.

only from the early pandemic period and it is helpful to review the context in which they were conducted and consider their generalizability to later time periods.

**Sensitivity analyses for the ICD-10-CM classification of nicotine dependence.** S5 Table reports the results of sensitivity analyses for the ICD-10-CM based definition of the exposure. The association of nicotine dependence with

mortality remained significant on defining the exposure as nicotine dependence, cigarettes. This is valuable as it allows for a better comparison of our study to the previous studies that examined the association of smoking with mortality.

**Sensitivity analyses for unmeasured confounders.** The E-value for the association of current *vs* never nicotine dependence with in-hospital mortality for the point estimate was 1.4777 and for the upper limit of the 95% confidence interval was 1.3401. Thus, unmeasured confounders would have to be associated on the risk ratio scale (beyond the adjusted for covariates) with both nicotine dependence and in-hospital mortality at this level to explain away the measured effect.

**Sensitivity analyses for selection biases.** In the present case of inference within a sampled population, the minimum level for selection bias, on the risk ratio scale, needed to explain away the observed effect of nicotine dependence on mortality has the same formula as the E-value for unmeasured confounding. Thus, if selection bias explains away the observed association, selection bias must result in an unmeasured variable that increases the risk of death by 1.3401 on the risk ratio scale and also differs between never and current dependents by 1.3401, but not less, on the risk-ratio scale.

**Exploratory analyses for selection biases.** From S6 Table, it can be seen that the examined diagnoses did not have lower crude in-hospital and short-term mortality compared to unexposed with most of the aRR also being non-significant or significantly greater than one. Thus, although the TIPUDF does not allow for determining severity of COVID-19 illness or differences in access to or processes of hospital care that may differ between never and currently nicotine dependent individuals and we consequently are not able to rule out selection bias, the results of our exploratory analyses do not support the presence of a material collider bias in our cohort derivation that explains away the measured effect of nicotine dependence on mortality.

**Subgroup analyses.** Table 6 and Table 7 summarize the results of subgroup analyses for the comparison of mortality between currently and never nicotine dependent individuals. Both crude in-hospital and short-term mortality rates were lower for currently nicotine dependent individuals compared to never nicotine dependent individuals across every examined subgroup. Most subgroups had statistically significant aRR < 1 and no subgroups had statistically significant aRR > 1. The results of subgroup analyses for in-hospital mortality are displayed as a forest plot in Fig 5.

## Discussion

### Key findings

In this population-based cohort study, critically-ill currently, formerly, and never nicotine dependent individuals with a principal diagnosis of COVID-19 had significantly different in-hospital mortality rates (10.7% *vs* 13.8% *vs* 16.1% respectively). This surprising observation, given the well-known negative impacts of nicotine dependence and the higher age and burden of co-morbid conditions among the nicotine dependent individuals, sparked our curiosity (Table 1). On risk adjusted analysis, current nicotine dependence was linked to a 10.5% lower risk of in-hospital mortality and a 10.7% lower risk of short-term mortality compared to never nicotine dependent individuals (Table 3). Formerly nicotine dependent individuals were much older than never nicotine dependent individuals but also had a reduced risk of in-hospital mortality compared to never nicotine dependent individuals. These findings challenge the existing understanding of the relationship between nicotine dependence and COVID-19 mortality, and emphasize the need for further research to fully comprehend the complex interplay of factors influencing COVID-19 outcomes.

An important aspect of our study has been the sensitivity analyses summarized in Table 4. We conducted unadjusted and risk adjusted analyses for both in-hospital and short-term mortality, which consistently showed lower risk for currently nicotine dependent individuals. These analyses were performed after restricting the data to hospitalizations aged ≥ 65 years, those using invasive mechanical ventilation, those with ≥ 3 organ dysfunctions, and those with Deyo comorbidity index ≥ 3. On subgroup analysis, crude in-hospital and short-term mortality rates were lower in currently nicotine dependent individuals compared to never nicotine dependent individuals across all examined hospitalization strata. The point estimates for the aRR were all less than one and a majority of the examined hospitalization strata had statistically significant aRR.

**Table 6. Subgroup analyses for the association of nicotine dependence and in-hospital mortality among critically-ill COVID-19 patients.**

| Subgroup | Current dependent | Never dependent | aRR [95% CI][a] | aRD [95% CI][b] | *p* value |
|---|---|---|---|---|---|
| | *in-hospital mortalities/ total no. (%)* | | | | |
| **All hospitalizations** | 1,115/ 10,452 (10.7) | 17,385/ 107,922 (16.1) | 0.8955 [0.8572 to 0.9356] | −0.0134 [−0.0185 to −0.0083] | < 0.0001 |
| **Sex** | | | | | |
| Male | 630/ 5,078 (12.4) | 9,340/ 52,351 (17.8) | 0.8904 [0.8433 to 0.9400] | −0.0161 [−0.0232 to −0.0089] | < 0.0001 |
| Female | 300/ 3,384 (8.9) | 7,461/ 52,232 (14.3) | 0.8980 [0.8227 to 0.9802] | −0.0107 [−0.0190 to −0.0024] | 0.0160 |
| **Age, years** | | | | | |
| 18–44 | 112/ 1,805 (6.2) | 1,352/ 16,431 (8.2) | 0.8866 [0.7789 to 1.0092] | −0.0082 [−0.0166 to 0.0003] | 0.0686 |
| 45–64 | 484/ 4,401 (11.0) | 5,746/ 38,453 (14.9) | 0.9771 [0.9163 to 1.0421] | −0.0028 [−0.0107 to 0.0050] | 0.4812 |
| ≥ 65 | 519/ 4,246 (12.2) | 10,287/ 53,038 (19.4) | 0.8153 [0.7639 to 0.8702] | −0.0299 [−0.0387 to −0.0211] | < 0.0001 |
| **Race** | | | | | |
| White | 557/ 5,575 (10.0) | 7,716/ 48,450 (15.9) | 0.8501 [0.7975 to 0.9060] | −0.0193 [−0.0264 to −0.0121] | < 0.0001 |
| Hispanic | 337/ 2,440 (13.8) | 6,705/ 37,731 (17.8) | 0.9569 [0.8908 to 1.0279] | −0.0065 [−0.0168 to 0.0039] | 0.2274 |
| Black | 147/ 1,841 (8.0) | 1,726/ 13,289 (13.0) | 0.8455 [0.7401 to 0.9659] | −0.0157 [−0.0275 to −0.0039] | 0.0135 |
| Other | 74/ 596 (12.4) | 1,238/ 8,450 (14.7) | 0.9937 [0.8485 to 1.1636] | −0.0008 [−0.0206 to 0.0190] | 0.9371 |
| **Deyo index** | | | | | |
| 0 | 351/ 3,484 (10.1) | 6,860/ 50,880 (13.5) | 0.9590 [0.8937 to 1.0291] | −0.0045 [−0.0119 to 0.0029] | 0.2445 |
| 1, 2 | 416/ 3,805 (10.9) | 5,893/ 33,723 (17.5) | 0.8937 [0.8328 to 0.9591] | −0.0142 [−0.0227 to −0.0056] | 0.0018 |
| ≥ 3 | 348/ 3,163 (11.0) | 4,632/ 23,319 (19.9) | 0.8219 [0.7550 to 0.8948] | −0.0267 [−0.0376 to −0.0159] | < 0.0001 |
| **Organ dysfunctions** | | | | | |
| 0, 1 | 276/ 6,913 (4.0) | 3,683/ 66,240 (5.6) | 0.8662 [0.7811 to 0.9607] | −0.0064 [−0.0108 to −0.0020] | 0.0065 |
| 2, 3 | 645/ 3,182 (20.3) | 10,254/ 36,441 (28.1) | 0.8872 [0.8386 to 0.9386] | −0.0272 [−0.0395 to −0.0150] | < 0.0001 |
| ≥ 4 | 194/ 357 (54.3) | 3,448/ 5,241 (65.8) | 0.9237 [0.8473 to 1.0070] | −0.0462 [−0.0951 to 0.0027] | 0.0716 |

[a] aRR [95% CI]: adjusted risk ratio and 95% confidence interval.

[b] aRD [95% CI]: adjusted risk difference and 95% confidence interval.

The decreasing crude and short-term mortalities over time presented in Table 5 have important implications for the interpretation of previous studies on the association of current nicotine dependence and mortality among COVID-19 patients. Mortality rates have fallen approximately four-fold since 2020. If similar patterns exist in other jurisdictions, then it is important to consider whether or not early studies on nicotine dependence among COVID-19 patients would lead to the same conclusions if repeated currently.

The results presented in S5 Table provide important details on the impact of cigarette smoking on mortality. First, it is seen that currently nicotine dependent individuals who smoke cigarettes compose the majority of the currently nicotine dependent individuals. Second, currently nicotine dependent individuals who smoke cigarettes had both the lowest crude in-hospital and short-term mortality rates amongst the examined subgroups of currently nicotine dependent individuals. This is important because it allows for a comparison of this study to previous reports about smoking and COVID-19.

Table 6 reports the expected finding that mortality rates increased with each of age, number of organ dysfunctions, and Deyo comorbidity index, and were also higher for males with each pattern observed for both currently and never nicotine dependent individuals. Table 6 also shows that Blacks had significantly lower crude mortality rates than Whites, Hispanics, and other races for both currently and never nicotine dependent individuals. This interesting observation is surprising and could be the subject of additional investigations. One potential explanation for these findings could be the influence of socio-economic factors, which may interact with nicotine dependence status and COVID-19 outcomes in complex ways.

Our exploratory and sensitivity analyses for selection bias are important because the TIPUDF does not allow for determining the severity of COVID-19 illness. As reported in S6 Table, other diagnoses do not demonstrate both the same

**Table 7. Subgroup analyses for the association of nicotine dependence and short-term mortality among critically-ill COVID-19 patients.**

| Subgroup | Current dependent | Never dependent | aRR [95% CI][a] | aRD [95% CI][b] | *p* value |
|---|---|---|---|---|---|
| | *short-term mortalities/ total no. (%)* | | | | |
| **All hospitalizations** | 1,410/ 10,452 (13.5) | 21,618/ 107,922 (20.0) | 0.8926 [0.8618 to 0.9245] | −0.0174 [−0.0225 to −0.0122] | < 0.0001 |
| **Sex** | | | | | |
| Male | 755/ 5,078 (14.9) | 11,272/ 52,351 (21.5) | 0.8683 [0.8304 to 0.9079] | −0.0237 [−0.0307 to −0.0166] | < 0.0001 |
| Female | 419/ 3,384 (12.4) | 9,632/ 52,232 (18.4) | 0.9434 [0.8823 to 1.0087] | −0.0078 [−0.0166 to 0.0010] | 0.0879 |
| **Age, years** | | | | | |
| 18–44 | 115/ 1,805 (6.4) | 1,393/ 16,431 (8.5) | 0.8761 [0.7748 to 0.9908] | −0.0094 [−0.0177 to −0.0010] | 0.0350 |
| 45–64 | 543/ 4,401 (12.3) | 6,262/ 38,453 (16.3) | 0.9535 [0.9012 to 1.0089] | −0.0066 [−0.0142 to 0.0011] | 0.0986 |
| ≥ 65 | 752/ 4,246 (17.7) | 13,963/ 53,038 (26.3) | 0.8418 [0.8029 to 0.8826] | −0.0354 [−0.0445 to −0.0263] | < 0.0001 |
| **Race** | | | | | |
| White | 758/ 5,575 (13.6) | 10,236/ 48,450 (21.1) | 0.8595 [0.8185 to 0.9026] | −0.0240 [−0.0314 to −0.0167] | < 0.0001 |
| Hispanic | 381/ 2,440 (15.6) | 7,758/ 37,731 (20.6) | 0.9505 [0.8939 to 1.0106] | −0.0085 [−0.0186 to 0.0016] | 0.1048 |
| Black | 182/ 1,841 (9.9) | 2,064/ 13,289 (15.5) | 0.8414 [0.7555 to 0.9372] | −0.0201 [−0.0319 to −0.0082] | 0.0017 |
| Other | 89/ 596 (14.9) | 1,560/ 8,450 (18.5) | 0.9985 [0.8720 to 1.1435] | −0.0002 [−0.0208 to 0.0204] | 0.9832 |
| **Deyo index** | | | | | |
| 0 | 381/ 3,484 (10.9) | 7,688/ 50,880 (15.1) | 0.9388 [0.8813 to 1.0001] | −0.0074 [−0.0147 to −0.0002] | 0.0502 |
| 1, 2 | 511/ 3,805 (13.4) | 7,464/ 33,723 (22.1) | 0.8862 [0.8364 to 0.9390] | −0.0188 [−0.0274 to −0.0102] | < 0.0001 |
| ≥ 3 | 518/ 3,163 (16.4) | 6,466/ 23,319 (27.7) | 0.8514 [0.8013 to 0.9046] | −0.0315 [−0.0428 to −0.0202] | < 0.0001 |
| **Organ dysfunctions** | | | | | |
| 0, 1 | 402/ 6,913 (5.8) | 5,332/ 66,240 (8.0) | 0.8646 [0.7987 to 0.9359] | −0.0095 [−0.0144 to −0.0046] | 0.0003 |
| 2, 3 | 786/ 3,182 (24.7) | 12,435/ 36,441 (34.1) | 0.8864 [0.8472 to 0.9274] | −0.0333 [−0.0452 to −0.0213] | < 0.0001 |
| ≥ 4 | 222/ 357 (62.2) | 3,851/ 5,241 (73.5) | 0.9517 [0.8903 to 1.0174] | −0.0325 [−0.0757 to 0.0106] | 0.1460 |

[a] aRR [95% CI]: adjusted risk ratio and 95% confidence interval.

[b] aRD [95% CI]: adjusted risk difference and 95% confidence interval

unadjusted and adjusted protective effect as nicotine dependence. Selection biases that may have resulted in an unmeasured confounder, such as illness severity, and a spurious protective effect for nicotine dependence did not do the same for these other diagnoses. However, it should be observed that from Table 1, nicotine dependent individuals were more often Black and less often Hispanic compared to never nicotine dependent individuals. This, combined with the lower mortality among Blacks and higher mortality among Hispanics seen in Table 6, suggests that socio-economic factors may have resulted in a selection bias. The E-value and the lower bound on selection bias allow for interpreting the extent of unmeasured confounding and selection bias needed to explain away the main results.

## Relationship to previous studies

Our study contributes to existing research through a comprehensive population-based cohort study of all hospitalizations with a principal diagnosis of COVID-19 admitted to the ICU at all acute care hospitals in Texas (excluding military facilities). This approach allowed for a thorough review of a large set of consecutive and unselected hospitalizations over the entirety of an expansive, contiguous, and diverse region during an extensive time period that extended significantly beyond the initial year of the COVID-19 pandemic. A few large-scale registry-based studies have concluded that nicotine dependent individuals have similar risk of in-hospital mortality compared to never nicotine dependent individuals. However, the existing literature has contradicting results and it appears that formerly nicotine dependent individuals have a higher risk of in-hospital mortality when compared to never and currently nicotine dependent individuals.

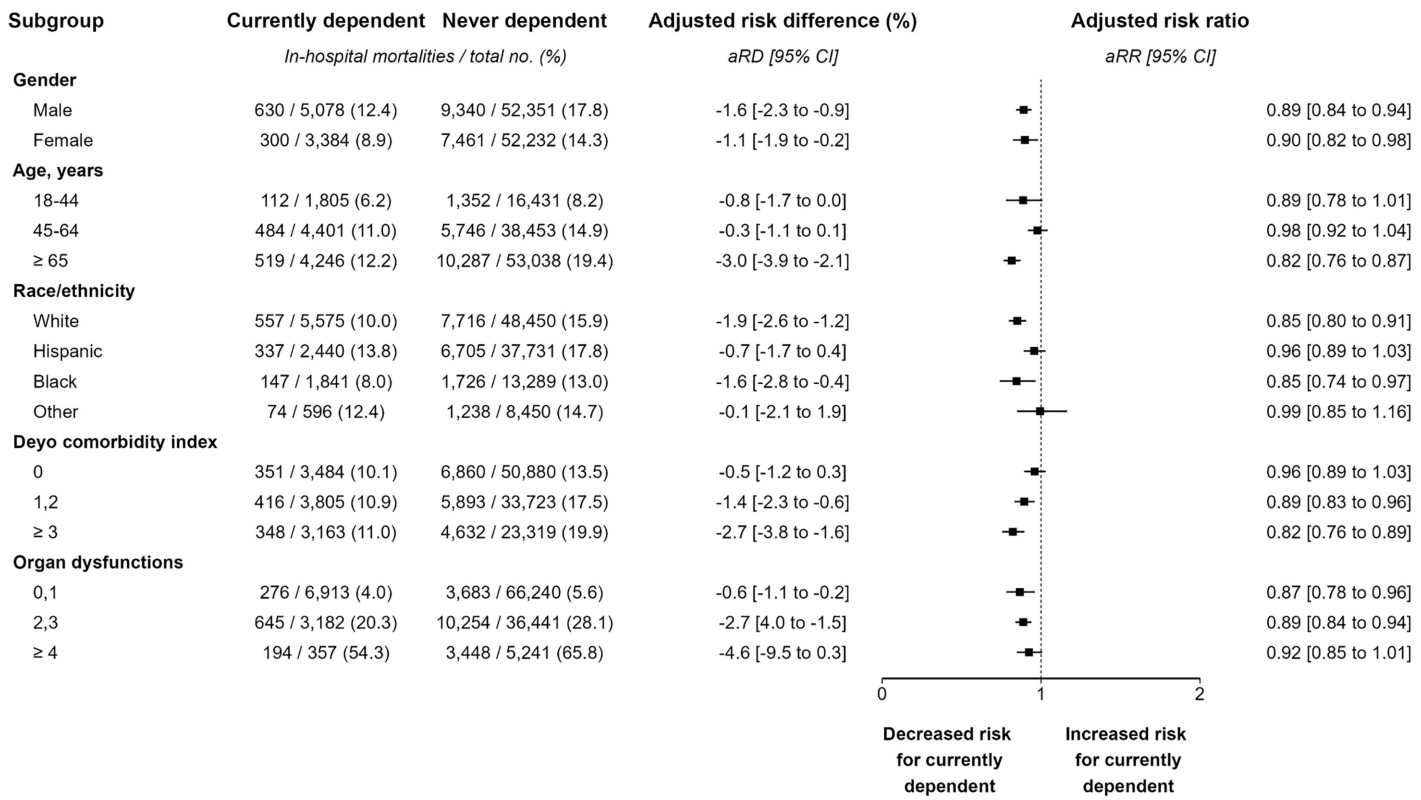

| Subgroup | Currently dependent | Never dependent | Adjusted risk difference (%) | | Adjusted risk ratio |
|---|---|---|---|---|---|
| | In-hospital mortalities / total no. (%) | | aRD [95% CI] | | aRR [95% CI] |
| **Gender** | | | | | |
| Male | 630 / 5,078 (12.4) | 9,340 / 52,351 (17.8) | -1.6 [-2.3 to -0.9] | | 0.89 [0.84 to 0.94] |
| Female | 300 / 3,384 (8.9) | 7,461 / 52,232 (14.3) | -1.1 [-1.9 to -0.2] | | 0.90 [0.82 to 0.98] |
| **Age, years** | | | | | |
| 18-44 | 112 / 1,805 (6.2) | 1,352 / 16,431 (8.2) | -0.8 [-1.7 to 0.0] | | 0.89 [0.78 to 1.01] |
| 45-64 | 484 / 4,401 (11.0) | 5,746 / 38,453 (14.9) | -0.3 [-1.1 to 0.1] | | 0.98 [0.92 to 1.04] |
| ≥ 65 | 519 / 4,246 (12.2) | 10,287 / 53,038 (19.4) | -3.0 [-3.9 to -2.1] | | 0.82 [0.76 to 0.87] |
| **Race/ethnicity** | | | | | |
| White | 557 / 5,575 (10.0) | 7,716 / 48,450 (15.9) | -1.9 [-2.6 to -1.2] | | 0.85 [0.80 to 0.91] |
| Hispanic | 337 / 2,440 (13.8) | 6,705 / 37,731 (17.8) | -0.7 [-1.7 to 0.4] | | 0.96 [0.89 to 1.03] |
| Black | 147 / 1,841 (8.0) | 1,726 / 13,289 (13.0) | -1.6 [-2.8 to -0.4] | | 0.85 [0.74 to 0.97] |
| Other | 74 / 596 (12.4) | 1,238 / 8,450 (14.7) | -0.1 [-2.1 to 1.9] | | 0.99 [0.85 to 1.16] |
| **Deyo comorbidity index** | | | | | |
| 0 | 351 / 3,484 (10.1) | 6,860 / 50,880 (13.5) | -0.5 [-1.2 to 0.3] | | 0.96 [0.89 to 1.03] |
| 1,2 | 416 / 3,805 (10.9) | 5,893 / 33,723 (17.5) | -1.4 [-2.3 to -0.6] | | 0.89 [0.83 to 0.96] |
| ≥ 3 | 348 / 3,163 (11.0) | 4,632 / 23,319 (19.9) | -2.7 [-3.8 to -1.6] | | 0.82 [0.76 to 0.89] |
| **Organ dysfunctions** | | | | | |
| 0,1 | 276 / 6,913 (4.0) | 3,683 / 66,240 (5.6) | -0.6 [-1.1 to -0.2] | | 0.87 [0.78 to 0.96] |
| 2,3 | 645 / 3,182 (20.3) | 10,254 / 36,441 (28.1) | -2.7 [4.0 to -1.5] | | 0.89 [0.84 to 0.94] |
| ≥ 4 | 194 / 357 (54.3) | 3,448 / 5,241 (65.8) | -4.6 [-9.5 to 0.3] | | 0.92 [0.85 to 1.01] |

Decreased risk for currently dependent | Increased risk for currently dependent

**Fig 5. Forest plot for in-hospital mortality.**

A study by Poudel *et al*, based on the American Heart Association's Get-With-The-Guidelines COVID-19 Registry, found that when compared to never nicotine dependent individuals, currently nicotine dependent individuals have a higher in-hospital mortality rate regardless of demographic and socioeconomic characteristics [36]. Their cohort was determined after matching currently nicotine dependent and never nicotine dependent individuals. The selection of hospitalizations in their study leads to differences in characteristics, for example their currently nicotine dependent cohort had higher rates of mechanical ventilation while our Table 1 shows current nicotine dependents having lower rates of mechanical ventilation. Another difference between our study and theirs is that 96.2% of the hospitalizations in their study occurred during 2020 while our study included a large number of hospitalizations through the end of 2024.

A study by Salah *et al*, showed that current nicotine dependence doubled the mortality among the COVID-19 patients based on meta-analysis of ten studies, where mortality also was higher among current dependents (29.4%) compared to never dependents (17.0%); risk ratio 2.07 (95% CI: 1.59 to 2.69) along with four studies (532 patients), that had no difference in mortality risk between current and former smokers (RR: 1.03; 95% CI 0.75 to 1.40) [37]. This study included only hospitalization during 2019 or 2020 and the target population for which their result applies is not clear. However, it is important to note that the study findings may be limited by the small sample size and the lack of clarity on the target population.

Hou *et al*, based on a global meta-analysis of 73 studies with a total of 863,313 COVID-19 patients, calculated a pooled relative risk of 1.19 (95% CI 1.12 to 1.27) and showed that nicotine dependence was a significant risk factor for mortality and confirmed the results through sensitivity analysis [38]. This global perspective, similar to the study by Salah *et al*,

underscores the broad applicability of their findings. The study was also based only on hospitalizations during the beginning of the COVID-19 pandemic, providing a comprehensive view but limited to the early stages of the disease.

Another meta-analysis by Patanavanich *et al*, concluded that currently nicotine dependent individuals are at higher mortality risk due to COVID-19 infection [39]. This study explored the association between nicotine dependence and COVID-19 mortality by systematically searching PubMed and Embase with 34 studies and 35,193 COVID-19 patients. The comprehensive nature of this study, including a large number of studies and patients, adds to its reliability. Random-effects meta-analysis confirmed that currently (OR 1.26, 95% CI 1.01 to 1.58) and formerly nicotine dependent individuals (OR 1.76, 95% CI 1.53 to 2.03) were associated with increased mortality risk. Further, the analysis also demonstrated that the mortality risk for currently nicotine dependent individuals remained consistent across age groups, whereas, for formerly nicotine dependent individuals, the risk decreased significantly with age. However, their study did not restrict to hospitalized ICU patients and included studies only during the first year of the pandemic.

Carolina Espejo-Paeres *et al* used a voluntary and international registry to select 5,224 hospitalizations for study and concluded that all-cause mortality was higher among either currently or formerly nicotine dependent individuals compared to never nicotine dependent individuals [40]. Their research differs from ours in that they focused on selected hospitalizations during the early months of the COVID-19 pandemic and utilized a relatively smaller sample size. Moreover, their study included only 307 smokers, with 61 deaths among that group, indicating a much smaller exposure group. In contrast, our research included a broader timeframe beyond the beginning of the pandemic and a much larger cohort.

Another study by Dessie *et al*, analyzed the pooled prevalence of mortality among patients with COVID-19 and showed that currently nicotine dependent individuals (OR = 1.42; 95% CI 1.01 to 1.83) had increased mortality [41]. This study was based on a systematic search of electronic databases, including 42 studies with a total of 423,117 hospitalizations, including all human studies, regardless of language, publication date, or region. Notably, the included studies were primarily published in 2020 and unlike our study, they did not restrict their focus to hospitalized patients.

A Brazilian national retrospective cohort study by Soares de Oliveira *et al*, examined adults hospitalized with COVID-19 using the SIVEP-Gripe database and concluded that nicotine dependence was associated with in-hospital mortality [42]. Although the study is a very large population-based study, only 2.1% of hospitalizations were identified as nicotine dependent, either currently or formerly nicotine dependent, compared to a total of 24.0% in our study. This eleven-fold difference in nicotine dependence rates between our study and theirs suggests that the identification of nicotine use through self-report in their study does not align with our definition using ICD-10-CM, which could potentially affect the comparability of results. Furthermore, mortality rates for never nicotine dependent individuals (31.8%) and currently nicotine dependent individuals (42.7%) in their study were much higher than those in our study, complicating direct comparisons.

Wilkinson LA *et al*, analyzed hospitalized veteran COVID-19-positive patients nationwide and assessed the association of nicotine dependence and mortality by multivariable logistic regression and the Cox Proportional Hazards model [43]. They concluded that mortality increased among formerly nicotine dependent individuals but found no significant difference in mortality between currently nicotine dependent individuals and never nicotine dependent individuals. In a different study by Razjouyan *et al*, using the same United States Veterans Health Administration (VHA) Corporate Data Wearhouse database, hospitalized formerly nicotine dependent individuals also had an increased risk of mortality compared to never nicotine dependent individuals and there was also no difference in the risk of in-hospital mortality between currently nicotine dependent individuals and never nicotine dependent individuals [44]. Our study excluded Veteran's Administration and military hospitals complicating comparisons between these studies and ours.

A UK Biobank prospective study by Prats-Uribe *et al* assessed a large cohort for the risk of COVID-19 infection and mortality among currently nicotine dependent individuals and never nicotine dependent individuals [45]. The study followed 402,978 participants in England from February 2020 to June 2020, linking data to hospital records and PCR tests. Results showed that older currently nicotine dependent individuals aged > 69 had significantly higher mortality from COVID-19 infection, with a risk ratio of 2.15 compared to never nicotine dependent individuals; the association was similar

among formerly nicotine dependent individuals and consistent across both genders. Additionally, currently nicotine dependent individuals under 69 were twice as likely to be infected with COVID-19 compared to never nicotine dependent individuals, with no difference among participants aged > 69. These findings highlight the vulnerability of older currently nicotine dependent individuals to severe COVID-19 outcomes and highlight the importance of targeted public health strategies, such as intensified preventive and cessation interventions, for this high-risk group.

Two other studies that used the same UK Biobank data reported conflicting results [46,47]. In Gao *et al*, when nicotine dependent individuals were compared to never nicotine dependent individuals, nicotine dependent individuals were at approximately one-third less risk of COVID-19 hospitalization, two-thirds lower risk for ICU admission, and one-fifth lower risk of death [46]. However, in Clift *et al*, a large-scale observational study using the same UK Biobank found nicotine dependence to be a casual effect in increasing the risk of severe COVID-19 outcomes [47]. In contrast to our study, these studies were not restricted to hospitalized patients.

Seyed Alinaghi *et al* completed an umbrella review of 27 systematic reviews synthesizing many studies varying by sample size, hospitalization status, geographical location, nicotine dependence definitions, facility type, and definition of mortality [48]. The majority of the systematic reviews agreed that nicotine dependence among COVID-19 had increased mortality rates along with higher disease severity, faster disease progression, higher hospitalization rates, more extended hospital stays, greater need for mechanical ventilation, and increased ICU admissions. This study contrasts with ours where we found that nicotine dependence had reduced need for mechanical ventilation compared to nicotine abstainers (9.5% *vs* 12.2%) and had shorter hospital length of stay (mean 8.4 days *vs* 9.3 days).

Piasecki *et al*, in a Wisconsin-based electronic health records study, found that formerly nicotine dependent individuals were at higher risk of in-hospital mortality compared to never nicotine dependent individuals [49]. They found that currently nicotine dependent individuals had a lower unadjusted mortality rate compared to never nicotine dependent individuals, but found no statistically significant association between current nicotine dependence and mortality on adjusted analysis. They also observed that nicotine replacement therapy reduced mortality for nicotine dependents.

A Living Rapid Review analyzed observational and experimental studies by Simons *et al*. on outcomes between nicotine dependence and COVID-19, including infection, hospitalization, disease severity, and mortality [50]. A total of 233 studies were considered, of which 32 were of good or fair quality and included in the meta-analyses. Currently nicotine dependent individuals had a reduced risk of COVID-19 infection compared to never nicotine dependent individuals, with a relative risk of 0.74. However, individuals who quit nicotine use showed an increased risk of hospitalization, greater disease severity, and higher mortality compared to never nicotine dependent individuals. For currently nicotine dependent individuals, the data were unclear regarding hospitalization and mortality, but there was a small association with increased disease severity. However, the study is not directly comparable since it is not restricted to hospitalized patients, and the target population to which the conclusion applies is not defined.

In summary, although many studies on the associations of nicotine use, COVID-19, and mortality have been published, they mostly differ from our study in their definitions of nicotine dependence, identification of nicotine dependent individuals, methods of nicotine use, definitions of mortality, the study periods considered, the criteria used in the selection of patients, and definition of a target population, complicating direct comparisons with our study. Previous studies have involved both hospitalized and non-hospitalized patients, only hospitalized patients, or only non-hospitalized patients. Nicotine dependence has been defined differently as either only currently nicotine dependent individuals, formerly nicotine dependent individuals, or both. Mortality has been defined as in-hospital mortality, mortality within 30 days of hospital discharge, or all mortality unrelated to hospitalization.

Critically, many studies included only patients from the beginning of the pandemic making their generalizability to later periods unclear and complicating comparison to later studies such as ours. Many previous studies have measured the average treatment effect in the combined population with other studies not clearly defining the target population to which their result applies. In contrast, our study has used an explicitly characterized overlap population and considers the direct effect within a population at clinical *equipoise*.

## Study strengths and limitations

This study has important strengths and limitations. Regarding strengths, the study investigates an interesting research question with few existing similar population-level studies. The study encompasses a large number of consecutive hospitalizations from a diverse population over five years, providing valuable insights into the impact of nicotine dependence on in-hospital and short-term mortality rates, which is crucial for understanding and improving public health outcomes. We followed the STROBE guidelines, used risk-adjusted analyses, subgroup analyses, and sensitivity analysis methods to limit confounders, explore the consistency of association, and check robustness. We quantified the sensitivity of our main result to unmeasured confounders using E-values and bounded the impact that selection biases must have to explain away the observed association. We demonstrated that the effective sample size of the overlap population remained comparable the original cohort while using a large and diverse collection of risk adjustment covariates.

This study, however, has important limitations that may affect the interpretation and generalizability of our findings. First, although we used the *CCSR for ICD-10-CM*, the *2024 ICD-10-CM Tabular List of Diseases and Injuries,* and the *FY 2024 ICD-10-CM Official Guidelines for Coding and Reporting* to assist in identifying ICD-10-CM codes for nicotine dependence, as presented in S1 Table and S2 Table, these codes have not been externally validated. Consequently, misclassification of nicotine dependence is possible. The existing variations among different hospitals in their protocols and interventions have a significant impact on the reported mortality rates among currently *vs* never nicotine dependent individuals. However, misclassification of nicotine dependence should be expected to reduce the observed differences between groups and thus lessen outcome differences between hospitalizations with different nicotine dependence statuses. Second, we could not identify repeated admissions which could not be accounted for within the risk-adjustment models. It should be noted, however, that similar methodological limitations affect epidemiological studies in the United States based on other deidentified administrative data sets, including the National Inpatient Sample (NIS). Third, the TIPUDF does not provide information on the severity of illness and the TIPUDF does not include detailed information on the delivery of healthcare and medication management that may be different for patients with different nicotine dependence statuses. Thus, residual confounding in our models remains a possible problem. Fourth, our study has not differentiated among nicotine dependence on the intensity of nicotine use or the time since cessation. Thus, our results should be interpreted only as a simplified summary that does not allow for more granular effects based on nicotine dependence intensity or time whose magnitudes and even directions may be different than we report. Fifth, ICD-10-CM codes do not allow for separating the direct effect of nicotine dependence on mortality from the indirect effect of nicotine dependence through tobacco smoke on mortality, complicating the interpretation of results.

Lastly, the generalizability of our findings to regions outside of Texas and to future time periods is unknown. Our review of prior research and our sensitivity analyses for year of discharge suggest that the association of nicotine dependence and mortality could change over time or vary by geographic region. The above are important limitations that affect the explanation and generalizability of our conclusions. Understanding these limitations is crucial for critically appraising the research and its potential impact on epidemiology and understanding of health outcomes.

## Conclusions

Hospitalized nicotine dependent individuals with a principal diagnosis of COVID-19 tended to be older, more frequently male, and exhibited a greater burden of co-morbid conditions compared to never nicotine dependent individuals. Surprisingly, despite these factors that would typically be associated with worse outcomes, nicotine dependent individuals demonstrated significantly lower crude and risk-adjusted in-hospital and short-term mortality rates than their never nicotine dependent counterparts. This unexpected finding stands in contrast to the outcomes reported in previous studies, highlighting a potentially complex relationship between nicotine dependence and COVID-19 mortality.

## Supporting information

**S1 Table. ICD-10-CM codes.**
(PDF)

**S2 Table. The distribution of nicotine dependence and tobacco use among critically-ill COVID-19 patient.**
(PDF)

**S3 Table. Variance inflation factors.**
(PDF)

**S4 Table. Alternative modeling for potential impact of multicollinearity.**
(PDF)

**S5 Table. Sensitivity analyses for type of current nicotine dependence or tobacco use.**
(PDF)

**S6 Table. Exploratory analyses for evaluation of potential selection bias.**
(PDF)

**S7 Table. Non association of covariates with nicotine dependence in the overlap weighted population.**
(PDF)

**S1 Fig. Density plots of propensity scores.**
(PDF)

**S1 File. R code for analysis.**
(PDF)

## Author contributions

**Conceptualization:** John Garza, Vani Selvan.

**Data curation:** John Garza.

**Formal analysis:** John Garza, Vani Selvan.

**Methodology:** John Garza, Vani Selvan.

**Project administration:** John Garza, Vani Selvan.

**Resources:** John Garza, Vani Selvan.

**Software:** John Garza.

**Supervision:** Vani Selvan.

**Validation:** John Garza.

**Visualization:** John Garza.

**Writing – original draft:** John Garza, Roy Sebastian, Brinkley Cover, Vani Selvan.

**Writing – review & editing:** John Garza, Roy Sebastian, Asley Sanchez, Vani Selvan.

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
