## [Editor Report · Decision Letter 0]

18 Oct 2024

Dear Dr. Garza,

Thank you for submitting your manuscript to PLOS ONE. After careful consideration, we feel that it has merit but does not fully meet PLOS ONE’s publication criteria as it currently stands. Therefore, we invite you to submit a revised version of the manuscript that addresses the points raised during the review process.

The manuscript contains several grammatical errors. The Introduction is weak and needs improvement. Furthermore, the Discussion relies heavily on potential mechanisms from other studies rather than focusing on the interpretation of the current study’s findings. A strong Discussion should integrate data from other studies while primarily emphasizing the interpretation of the authors' own results. Currently, there is an overemphasis on external studies and insufficient interpretation and discussion of the authors' own work. Please address these issues before resubmitting for peer review.

We look forward to receiving your revised manuscript.

Kind regards,

Usama Waqar, M.B.B.S

Academic Editor

PLOS ONE

Journal Requirements:

Nicotine inhibits the production of proinflammatory mediators in human monocytes by suppression of I-κB phosphorylation and nuclear factor-κB transcriptional activity through nicotinic acetylcholine receptor α7 - https://doi.org/10.1111/j.1365-2249.2006.03169.x

(among others)

In your revision ensure you cite all your sources (including your own works), and quote or rephrase any duplicated text outside the methods section. Further consideration is dependent on these concerns being addressed.

---

## [Author Response · Author response to Decision Letter 1]

26 Nov 2024

Dear PLOS ONE,

Thank you for the reviewer comments regarding the submission:

“A paradoxical outcome of smokers among hospitalized patients with COVID-19: A population-based cohort study”

(PONE-D-24-31829).

A point-by-point response to the reviewer comments is provided below.

• “The manuscript contains several grammatical errors.”

We have corrected many grammatical errors and have carefully reviewed the manuscript for grammatical

and stylistic improvements. For example, the abstract of the attached file “Revised Manuscript with Track

Changes” has new text highlighted in green and deletions in red with strikethrough.

• “The Introduction is weak and needs improvement.”

The introduction has been redone. There is a different emphasis with new references added. We have

attempted to make the introduction align with the scale and scope of similar manuscripts on related topics.

• “Furthermore, the Discussion relies heavily on potential mechanisms from other studies rather than

focusing on the interpretation of the current study’s findings. A strong Discussion should integrate data

from other studies while primarily emphasizing the interpretation of the authors' own results. Currently,

there is an overemphasis on external studies and insufficient interpretation and discussion of the

authors' own work.”

After considering the reviewers comments, our research team decided to remove the original discussion of

potential mechanisms from the manuscript. Although we believe hypotheses regarding potential

mechanisms are important and interesting, we also believe that the previous discussion is better suited for

a separate research project and is out of place here.

We have divided a new discussion section into two subsections. The new “key findings” subsection adds

additional commentary on tables one through four that we believe will aid readers in better interpreting the

study results. A new “relationship to previous studies” section highlights the differences in design and

conclusions between our study and previous studies on the topic. We believe it is worth mentioning that a

large and diverse volume of research has been published on this topic from authors spanning the globe,

making it impractical to mention all relevant studies.

---

## [Decision Letter · Decision Letter 1]

7 Jan 2025

Dear Dr. Garza,

Thank you for submitting your manuscript to PLOS ONE. After careful consideration, we feel that it has merit but does not fully meet PLOS ONE’s publication criteria as it currently stands. Therefore, we invite you to submit a revised version of the manuscript that addresses the points raised during the review process.

We look forward to receiving your revised manuscript.

Kind regards,

Usama Waqar, M.B.B.S

Academic Editor

PLOS ONE

Reviewers' comments:

Reviewer's Responses to Questions

**Comments to the Author**

Reviewer #1: All comments have been addressed

Reviewer #2: (No Response)

2. Is the manuscript technically sound, and do the data support the conclusions?

Reviewer #1: Yes

Reviewer #2: Yes

3. Has the statistical analysis been performed appropriately and rigorously?

Reviewer #1: Yes

Reviewer #2: Yes

4. Have the authors made all data underlying the findings in their manuscript fully available?

Reviewer #1: Yes

Reviewer #2: Yes

5. Is the manuscript presented in an intelligible fashion and written in standard English?

Reviewer #1: Yes

Reviewer #2: Yes

Reviewer #1: (No Response)

Reviewer #2: The authors have made changes as per the previous reviewer recommendations. It is an interesting study with a large patient population. Their results are different than many of the earlier observational and meta-analysis looking at smoking and COVID-19. This would have given an opportunity to have an interesting discussion and highlight their results and how they got strikingly different results. The attached discussion in the revised manuscript still lacks that and they discuss no more than 2-3 studies. I strongly suggest the authors to have an extensive discussion highlighting other meta-analysis and why do their results differ. An entire page on discussion is just summarizing the results.

Like for example- Some large-scale registry-based studies have concluded that current smokers have similar risk of in-hospital mortality compared to nonsmokers- No reference attached. How did these studies differ than theirs and why do they have different results- is it because of race, gender, age or geographical location?

Other registry-based studies have in contrast concluded that smokers are at increased risk of in-hospital mortality. Using

data from a voluntary American Heart Association hospital data registry, Poudel et al found smokers to be at increased risk of in-hospital mortality. Just mentioning results is not discussion- they need to then further elaborate why are their results different and how we should infer this study.

.

Reviewer #1: No

Reviewer #2: No

---

## [Author Response · Author response to Decision Letter 2]

21 Feb 2025

We thank the reviewer for these important comments.

We have increased the number of previous studies considered to fifteen.

We emphasize differences in the definitions of smoking, mortality, cohort, and time periods of the different studies.

Within the methods and materials section, we now explain how our exposure variable should be interpreted in relation to those used in other studies.

We state that differences in parameters make many previous studies incomparable to ours.

We emphasize the characteristics of our study that make it a meaningful contribution to what is known on the topic.

In the conclusions section, we conclude that a decisive consensus on the association of smoking with mortality requires additional research.

---

## [Decision Letter · Decision Letter 2]

17 Apr 2025

Dear Dr. Garza,

We look forward to receiving your revised manuscript.

Kind regards,

Usama Waqar, M.B.B.S

Academic Editor

PLOS ONE

Reviewers' comments:

Reviewer's Responses to Questions

**Comments to the Author**

Reviewer #3: (No Response)

Reviewer #4: All comments have been addressed

2. Is the manuscript technically sound, and do the data support the conclusions?

Reviewer #3: Yes

Reviewer #4: Yes

3. Has the statistical analysis been performed appropriately and rigorously?

Reviewer #3: Yes

Reviewer #4: Yes

4. Have the authors made all data underlying the findings in their manuscript fully available?

Reviewer #3: Yes

Reviewer #4: Yes

5. Is the manuscript presented in an intelligible fashion and written in standard English?

Reviewer #3: Yes

Reviewer #4: Yes

Reviewer #3: The authors have a retrospective large database analyses investigating the association between smoking and in-hospital mortality among COVID-19 patients. The study used a large cohort and used rigorous statistical methodologies. However, there are multiple comments the authors need to address:

Major Concerns:

1. Study Cohort Definition and Indication for Hospitalization

• The manuscript states that the study includes hospitalized patients with a diagnosis of COVID-19 (ICD-10-CM U07.1 in any diagnosis column). This means that the cohort includes both patients admitted due to COVID-19 and those hospitalized for other reasons who happened to have a COVID-19 diagnosis. This significantly affects the interpretation of the results. Patients who were hospitalized due to COVID-19 may have different mortality risks than those hospitalized for another condition (e.g., cancer, heart failure) but tested positive for COVID-19. Thus, smoking associations with COVID-19 related mortality can be skewed.

• While the authors attempted to adjust for this by including a sensitivity analysis limited to patients with a principal diagnosis of COVID-19, this approach does not fully resolve potential confounding.

• Recommendation: The authors should clarify in discussion how this affects the results and could explain the results of this study.

2. Potential Survivor Bias and “Smoker’s Paradox”

• The finding that smokers had lower in-hospital mortality contradicts prior literature suggesting that smoking is a risk factor for severe COVID-19 outcomes. This could be due to survivor bias, where:

o Sicker smokers (with severe chronic lung disease, cardiovascular disease) may have died before hospitalization, meaning only relatively healthier smokers were included in the dataset.

o More severe COVID-19 cases among smokers may have been less likely to make it to the hospital.

• Recommendation: The discussion should highlight and critically assess the potential survivor bias as a key limitation. If possible, authors should compare pre-hospital mortality rates among smokers and non-smokers (e.g., by referring to external epidemiological data).

3. Smoking Definition and Intensity

• Smoking status was defined using ICD-10-CM codes for current and former smokers (Table S1). However:

o How recent is “former smoking” considered?

o Was smoking intensity assessed (e.g., pack-years)? Prior literature has shown that smoking severity impacts disease outcomes.

• Recommendation: If possible, stratifying former smokers by time since cessation and current smokers by smoking intensity could provide more granular insights. Otherwise, the limitation of not differentiating smoking intensity should be discussed.

4. Inconsistencies in ICU Admissions, Mechanical Ventilation, and Disease Severity

• Non-smokers had higher ICU admission rates (53.1% vs. 50.6%) and required more mechanical ventilation (9.7% vs. 7.1%), despite having a lower Deyo comorbidity index. This is counterintuitive, as one would expect more critically ill smokers requiring these interventions.

• Recommendation: The authors should explore potential reasons for this discrepancy.

o Were non-smokers admitted later in the disease course when they were already severely ill?

o Were hospital admission and treatment practices different for smokers vs. non-smokers?

o If this discrepancy was due to unmeasured confounders, this should be explicitly stated in the discussion.

5. Model Complexity and Overfitting Risk

• The multivariable models included 11 covariates. While the dataset is large, including too many covariates in a logistic regression can introduce overfitting and multicollinearity issues.

• Recommendation:

o The authors assessed variance inflation factors (VIFs) for multicollinearity, its details should be shared.

Minor Concerns and Additional Recommendations:

6. Statistical Considerations

• Line 93-94: "Continuous variables are summarized as mean and standard deviation."

o Was normality tested before using mean (SD)? If data were non-normally distributed, median (IQR) should be reported instead.

• Line 94: “Fisher’s test was used for comparison of categorical variables.”

o Why was Fisher’s test used instead of chi-square? Fisher’s test is typically used when expected cell counts are small (<5). If used for large categorical variables, a chi-square test would be more appropriate.

7. Smoking and Hospital Disposition

• More smokers left against medical advice (LAMA) (2.4% vs. 1.2%).

o How were these patients accounted for? Could their outcomes have differed from hospitalized patients? It should be noted in the discussion.

• Insurance Data Missing for More Non-Smokers.

o Was there a difference in healthcare access? Did this lead to delayed hospital presentation or differences in treatment? This potential confounder should be acknowledged.

8. Limitations of the TIPUDF Database

• The authors acknowledge that TIPUDF does not provide illness severity data or detailed treatment information (e.g., oxygen therapy, corticosteroid use).

• Recommendation: Expand on this limitation in the discussion. The lack of disease severity data could lead to residual confounding, as smokers and non-smokers may have had different baseline COVID-19 severity that was not captured in the dataset.

9. Discussion of Prior Studies

• The discussion compares findings to multiple prior studies but does not critically assess key methodological differences in enough detail.

o Line 257: “The tenfold difference in smoking between our study and theirs suggests that smoking was defined and identified differently.”

What was their definition, and how does it differ?

o Line 225-229: "Many prior studies only included hospitalizations during the early pandemic (2019-2020)."

What are the implications? Earlier in the pandemic, ICU capacity was more limited, and treatment protocols were different.

Did later treatment strategies (e.g., corticosteroids, antivirals) modify the effect of smoking on mortality?

Reviewer #4: Changes looks good

-add about the deyo comorbidity index and how did you calculate in the methodology section

-Page 14: Restructure the sentence 'Thus, the association of smoking and mortality begins to fade away as the health of COVID-19 patients deteriorate' as this is the result based on your analysis and not a established fact.You can say in simple words that our analysis show diminishing association between smoking and mortality among severe covid-19 patients.

-your limitations should include a possibility of reporting bias among patients regarding their smoking habits. it should also include that both former and current smokers were grouped into one exposure categories. they were not analyzed as compared to previous studies.

.

Reviewer #3: No

Reviewer #4: No

---

## [Author Response · Author response to Decision Letter 3]

11 Sep 2025

We have attached a pdf of the responses to reviewers. This allows including several new tables and figures that help answer reviewer questions.

---

## [Decision Letter · Decision Letter 3]

8 Nov 2025

Dear Dr.  Garza,

We look forward to receiving your revised manuscript.

Kind regards,

Usama Waqar, M.B.B.S

Academic Editor

PLOS ONE

Journal Requirements:

Additional Editor Comments:

The methods section currently states that "hospitalizations with a diagnosis of both former and current smoking were excluded from the study (n = 25)". Considering that the authors compared never smokers with current and former smokers, it is unclear what the authors meant to indicate here. Please clarify.

Reviewer's Responses to Questions

**Comments to the Author**

Reviewer #3: All comments have been addressed

2. Is the manuscript technically sound, and do the data support the conclusions?

Reviewer #3: Yes

3. Has the statistical analysis been performed appropriately and rigorously?

Reviewer #3: Yes

4. Have the authors made all data underlying the findings in their manuscript fully available?

Reviewer #3: Yes

5. Is the manuscript presented in an intelligible fashion and written in standard English?

Reviewer #3: Yes

Reviewer #3: The authors have addressed the core methodological concerns with substantial and well-reasoned revisions: redefining the cohort to principal COVID-19 diagnosis, excluding AMA discharges, expanding limitations, separating smoking exposure into never/former/current, adding quantitative bias analyses, implementing propensity score overlap weighting with balance diagnostics, reporting VIFs with an alternate model (w/ and w/o Deyo), switching to χ² for large-sample categorical comparisons, and providing year-stratified sensitivity analyses. The additions (revised flow diagram, DAGs, overlap-weighted balance table) meaningfully improve internal validity and transparency.

.

Reviewer #3: No

---

## [Author Response · Author response to Decision Letter 4]

5 Dec 2025

November 10, 2025

Dear PLOS ONE,

We appreciate the editor comment and the positive comments by the reviewer. We have adjusted the manuscript by deleting the sentence that is ambiguous and confusing. Note that the same information is available in the cohort derivation diagram where the meaning is clearer in that it leads to a partition of hospitalizations with three mutually exclusive groups; never smokers, former smokers, and current smokers.

Sincerely,

John Garza, PhD

Associate Professor

Texas Tech University School of Medicine

gar31703@ttuhsc.edu

512-350-1444

• Additional Editor Comments:

The methods section currently states that "hospitalizations with a diagnosis of both former and current smoking were excluded from the study (n = 25)". Considering that the authors compared never smokers with current and former smokers, it is unclear what the authors meant to indicate here. Please clarify.

In our effort to divide hospitalizations into never smokers, former smokers, and current smokers, we used ICD-10-CM codes as presented in the Supplementary Materials. The reason for this sentence relates to information found on the American Lung Association Billing Guide for Tobacco Screening and Cessation 2021 (linked below)

https://www.lung.org/getmedia/275e15df-413d-450f-9bed-b98a9fb04e1a/ala-billing-guide-2021.pdf

The information found on pages nine and thirteen indicate that ICD-10-CM code Z87891 should not be used together with ICD-10-CM F17 codes. Thus, each hospitalization should either have only Z87891 and not F17 codes, have a F17 code but not Z87891, or not have Z87891 and not have F17 codes. This was mostly observed in our cohort derivation except for a small number (n=25) and we decided that we could not determine the correct smoking status of these hospitalizations. They were therefore excluded.

We agree with the editor’s assessment that the sentence in unclear and have chosen to delete this sentence from the manuscript. Figure 2 however still communicates this fact but it is clearer from the figure what is intended and the diagram ends in nonoverlapping groups. We respectfully request that the editor accept our proposed resolution.

6. Review Comments to the Author

Reviewer #3: The authors have addressed the core methodological concerns with substantial and well-reasoned revisions: redefining the cohort to principal COVID-19 diagnosis, excluding AMA discharges, expanding limitations, separating smoking exposure into never/former/current, adding quantitative bias analyses, implementing propensity score overlap weighting with balance diagnostics, reporting VIFs with an alternate model (w/ and w/o Deyo), switching to χ² for large-sample categorical comparisons, and providing year-stratified sensitivity analyses. The additions (revised flow diagram, DAGs, overlap-weighted balance table) meaningfully improve internal validity and transparency.

We thank the reviewer for the many helpful comments and the time invested in carefully checking the many details of the study.

---

## [Decision Letter · Decision Letter 4]

5 Feb 2026

Dear Dr. Garza,

Thank you for submitting your manuscript to PLOS ONE. After careful consideration, we feel that it has merit but does not fully meet PLOS ONE’s publication criteria as it currently stands. Therefore, we invite you to submit a revised version of the manuscript that addresses the points raised during the review process.

**The critical issue is that it appears that incorrect codes have been used. Please see the comments from the editor below and clarify, correct, or edit as appropriate. Please note that the analyses may be performed again using a different set of patients according to the correct codes or the conclusions (from the title on) must be thoroughly edited to clearly indicated that the population analyzed was nicotine users, not necessarily smokers.**

We look forward to receiving your revised manuscript.

Kind regards,

Luis M Schang, MV. Ph.D.

Section Editor

PLOS One

Journal Requirements:

Additional Editor Comments :

Thank you for addressing all previous comments. The manuscript has now been reviewed by two additional experts.

From the description provided in the supplementary materials and methods, it appears that individuals were classified as current smokers using the following codings:

F17200, F17201, F17203, F17208, F17209, F17210, F17211, F17213, F17218, and F17219.

However, all codes F1720X are used for Nicotine addiction, which includes smokers, chewing tobacco users, vaping, and nicotine gum or patch users.

Considering that there are specific codes referencing to smokers, the use of F1720X strongly suggests other types of addictions.

Only F1721X codes refer to smokers exclusively.

Moreover, F17211 (and F17201) refer to people in remission and are not to be used for current smokers.

Please clarify if the coding used was different from what appears to be described in the methods section. If the coding used was as described, though, the results must be re-analyzed using only the correct codes, or the conclusions (from the title on) must be changed from smokers/smoking to nicotine users / addiction and the inclusion of non-smokers must be clearly described.

If F17211 or F17201 was used for current smokers/ nicotine users, the data will need to be re-evaluated or the conclusions from the tile on must be changed to current or in remission nicotine users/smokers

Reviewers' comments:

Reviewer's Responses to Questions

**Comments to the Author**

Reviewer #5: All comments have been addressed

2. Is the manuscript technically sound, and do the data support the conclusions?

Reviewer #5: Yes

3. Has the statistical analysis been performed appropriately and rigorously?

Reviewer #5: Yes

4. Have the authors made all data underlying the findings in their manuscript fully available?

Reviewer #5: Yes

5. Is the manuscript presented in an intelligible fashion and written in standard English?

Reviewer #5: Yes

Reviewer #5: (No Response)

.

Reviewer #5: No

---

## [Author Response · Author response to Decision Letter 5]

6 Mar 2026

Dear PLOSONE,

Thank you for the insightful comments and the opportunity to revise the manuscript. In response to the editor’s valuable suggestions, we have made adjustments to the manuscript. After studying the ICD-10-CM Tabular List of Diseases and Injuries, the Clinical Classification Software Refined for ICD-10-CM Diagnosis, and the FY 2024 ICD-10-CM Official Guidelines for Coding and Reporting, we realized that both “smoking” and “nicotine user” are not well defined or clearly identifiable within ICD-10-CM. As a result, we have changed the exposure to be “nicotine dependence” which, in contrast, is well defined in ICD-10-CM. The methods and structure of the manuscript are the same as before and the results are practically the same as before. Key adjustments to the manuscript, intended to improve technical accuracy, assist readers in interpreting the results, and assist in evaluating the study, include

1. A revised Table S1 that using the editor’s guidance for ICD-10-CM definition of currently and formerly nicotine dependent individuals

2. A new Table S2 in the supplementary materials file that lists each individual ICD-10-CM code for nicotine dependence along with the code description and prevalence within the cohort

3. A new Table S6 in the supplementary materials file that provides a sensitivity analysis for the class of nicotine dependence including for nicotine dependence cigarettes

4. A comprehensive review of ICD-10-CM codes for smoking and nicotine user vs nicotine dependence in the exposure and outcome subsection of the materials and methods section

5. A revised and improved Figure 1 (Directed acyclic graph)

6. Expansion of the study cohort to include Q3 and Q4 of 2024 as this data was now available

7. Additional R code in the supplementary materials file demonstrating properties of clinical equipoise in the overlap weighted population

8. Additional R code in the supplementary materials file comparing the confidence intervals obtained through nonparametric bootstraps to those obtained using the primary analysis procedure.

9. Throughout the manuscript “smoker” has been replaced by “nicotine dependent individual”

10. We have added information on the effective sample size of the overlap weighted population

The critical issue is that it appears that incorrect codes have been used. Please see the comments from the editor below and clarify, correct, or edit as appropriate. Please note that the analyses may be performed again using a different set of patients according to the correct codes or the conclusions (from the title on) must be thoroughly edited to clearly indicated that the population analyzed was nicotine users, not necessarily smokers.

Additional Editor Comments :

Thank you for addressing all previous comments. The manuscript has now been reviewed by two additional experts.

We appreciate the time invested by the additional reviewers.

From the description provided in the supplementary materials and methods, it appears that individuals were classified as current smokers using the following codings:

F17200, F17201, F17203, F17208, F17209, F17210, F17211, F17213, F17218, and F17219.

However, all codes F1720X are used for Nicotine addiction, which includes smokers, chewing tobacco users, vaping, and nicotine gum or patch users.

Considering that there are specific codes referencing to smokers, the use of F1720X strongly suggests other types of addictions.

Only F1721X codes refer to smokers exclusively.

Moreover, F17211 (and F17201) refer to people in remission and are not to be used for current smokers.

Please clarify if the coding used was different from what appears to be described in the methods section. If the coding used was as described, though, the results must be re-analyzed using only the correct codes, or the conclusions (from the title on) must be changed from smokers/smoking to nicotine users / addiction and the inclusion of non-smokers must be clearly described.

If F17211 or F17201 was used for current smokers/ nicotine users, the data will need to be re-evaluated or the conclusions from the tile on must be changed to current or in remission nicotine users/smokers

Thank you for the insightful comments and the opportunity to revise the manuscript. In response to the editor’s valuable suggestions, we have made adjustments to the manuscript. After studying the ICD-10-CM Tabular List of Diseases and Injuries, the Clinical Classification Software Refined for ICD-10-CM Diagnosis, and the FY 2024 ICD-10-CM Official Guidelines for Coding and Reporting, we realized that both “smoking” and “nicotine user” are not well defined or clearly identifiable within ICD-10-CM. As a result, we have changed the exposure to be “nicotine dependence” which, in contrast, is well defined in ICD-10-CM. The methods and structure of the manuscript are the same as before and the results are practically the same as before. Key adjustments to the manuscript, intended to improve technical accuracy, assist readers in interpreting the results, and assist in evaluating the study, include

1. A revised Table S1 that using the editor’s guidance for ICD-10-CM definition of currently and formerly nicotine dependent individuals

2. A new Table S2 in the supplementary materials file that lists each individual ICD-10-CM code for nicotine dependence along with the code description and prevalence within the cohort

3. A new Table S6 in the supplementary materials file that provides a sensitivity analysis for the class of nicotine dependence including for nicotine dependence cigarettes

4. A comprehensive review of ICD-10-CM codes for smoking and nicotine user vs nicotine dependence in the exposure and outcome subsection of the materials and methods section

5. A revised and improved Figure 1 (Directed acyclic graph)

6. Expansion of the study cohort to include Q3 and Q4 of 2024 as this data was now available

7. Additional R code in the supplementary materials file demonstrating properties of clinical equipoise in the overlap weighted population

8. Additional R code in the supplementary materials file comparing the confidence intervals obtained through nonparametric bootstraps to those obtained using the primary analysis procedure.

9. Throughout the manuscript “smoker” has been replaced by “nicotine dependent individual”

10. We have added information on the effective sample size of the overlap weighted population

Sincerely,

John Garza, PhD

Associate Professor

Texas Tech University School of Medicine

---

## [Decision Letter · Decision Letter 5]

10 Mar 2026

Dear Dr. Garza,

Please refer to the editor's comments below for important suggestions about minor modifications that would improve the manuscript.

We look forward to receiving your revised manuscript.

Kind regards,

Luis M Schang, MV. Ph.D.

Section Editor

PLOS One

Journal Requirements:

Additional Editor Comments:

Thank you for submitting the revised manuscript, which has been enriched by the discussion about the potential confounders in the classification of patients. On reading the revised manuscript, it is not obvious why the title still includes the world "paradoxical". The manuscript makes it clear now that this study properly classifies the patients, which makes it difficult to compare to other published data. Not sure any adjective is needed, but "unpredicted" appears more consistent with the actual manuscript if one is to be used.

The authors may want to consider including a brief discussion that the classification used does not include all smokers, many of whom are not dependent/addict to nicotine. It would help raise the awareness of the readers of any article analyzing the effects of "smoking" or nicotine addiction/dependence, or even use on any morbidity.

Reviewers' comments:

Reviewer's Responses to Questions

**Comments to the Author**

Reviewer #5: All comments have been addressed

2. Is the manuscript technically sound, and do the data support the conclusions?

Reviewer #5: (No Response)

3. Has the statistical analysis been performed appropriately and rigorously?

Reviewer #5: (No Response)

4. Have the authors made all data underlying the findings in their manuscript fully available?

Reviewer #5: (No Response)

5. Is the manuscript presented in an intelligible fashion and written in standard English?

Reviewer #5: (No Response)

Reviewer #5: (No Response)

.

Reviewer #5: No

---

## [Author Response · Author response to Decision Letter 6]

13 Mar 2026

March 11, 2026

Dear PLOSONE,

Thank you for the helpful editor comments. We have revised the manuscript and believe the revisions increase the accuracy and interpretability of the study. The title of the manuscript has been changed to “Nicotine dependence among critically ill COVID-19 patients: A population-based cohort study.” We have added an additional paragraph in the introduction section emphasizing that smoking and nicotine dependence are not subsets of each other preventing a direct comparison of this study to others using smoking as the exposure.

Sincerely,

John Garza, PhD

Associate Professor

Texas Tech University School of Medicine

gar31703@ttuhsc.edu

---

## [Editor Report · Decision Letter 6]

17 Mar 2026

Nicotine dependence among critically ill COVID-19 patients: A population-based cohort study

PONE-D-24-31829R6

Dear Dr. Garza,

We’re pleased to inform you that your manuscript has been judged scientifically suitable for publication and will be formally accepted for publication once it meets all outstanding technical requirements.

Kind regards,

Luis M Schang, MV. Ph.D.

Section Editor

PLOS One

Additional Editor Comments (optional):

Thank you very much for the prompt resubmission including the suggested modifications.

I am of the opinion that the fully revised manuscript now makes a major contribution to the body of scientific literature in the field. Congratulations on the excellent work and your willingness to receive feedback and improve the manuscript based on it.
---

## [Editor Report · Acceptance letter]

PONE-D-24-31829R6

PLOS One

Dear Dr. Garza,

I'm pleased to inform you that your manuscript has been deemed suitable for publication in PLOS One. Congratulations! Your manuscript is now being handed over to our production team.

Kind regards,

on behalf of

Dr. Luis M Schang

Section Editor

PLOS One